# Solving General-Utility Markov Decision Processes in the Single-Trial Regime with Online Planning

**Pedro P. Santos**
INESC-ID & Instituto Superior Técnico
Lisbon, Portugal
`pedro.pinto.santos@tecnico.ulisboa.pt`

**Alberto Sardinha**
PUC-Rio
Rio de Janeiro, Brazil
`sardinha@inf.puc-rio.br`

**Francisco S. Melo**
INESC-ID & Instituto Superior Técnico
Lisbon, Portugal
`fmelo@inesc-id.pt`

## Abstract

In this work, we contribute the first approach to solve infinite-horizon discounted general-utility Markov decision processes (GUMDPs) in the single-trial regime, i.e., when the agent's performance is evaluated based on a single trajectory. First, we provide some fundamental results regarding policy optimization in the single-trial regime, investigating which class of policies suffices for optimality, casting our problem as a particular MDP that is equivalent to our original problem, as well as studying the computational hardness of policy optimization in the single-trial regime. Second, we show how we can leverage online planning techniques, in particular a Monte-Carlo tree search algorithm, to solve GUMDPs in the single-trial regime. Third, we provide experimental results showcasing the superior performance of our approach in comparison to relevant baselines.

## 1 Introduction

Markov decision processes (MDPs) have found a wide range of applications in different domains such as inventory management (Dvoretzky et al., 1952), queueing control (Stidham, 1978), or optimal stopping (Chow et al., 1971). MDPs are also of key importance in the field of reinforcement learning (RL) Sutton & Barto (2018), where the agent-environment interaction is usually modeled by resorting to the framework of MDPs. In addition, recent years have seen significant progress in applying RL techniques to different domains (Mnih et al., 2015; Silver et al., 2017; Lillicrap et al., 2016), attesting to the flexibility of the MDP framework with respect to objective-specification.

However, despite providing a flexible framework concerning objective-specification, previous research has shown that multiple relevant objectives cannot be easily expressed within the MDP framework (Abel et al., 2022). Such objectives include, but not limited to, imitation learning (Hussein et al., 2017; Osa et al., 2018), pure exploration problems (Hazan et al., 2019), risk-averse RL (García et al., 2015), diverse skills discovery (Eysenbach et al., 2018; Achiam et al., 2018), constrained MDPs (Altman, 1999; Efroni et al., 2020), and adversarial MDPs (Rosenberg & Mansour, 2019). All aforementioned objectives can be cast under the framework of general-utility Markov decision processes (GUMDPs) (Santos et al., 2024). GUMDPs generalize the framework of MDPs by allowing the objective to be a non-linear function of the occupancy (the frequency of visitation of state-action pairs induced when running a given policy on the MDP). Recent works unified such objectives under the GUMDP framework and proposed algorithms to solve GUMDPs with convex objectives (Zhang et al., 2020; Geist et al., 2022; Zahavy et al., 2021).

Unfortunately, in GUMDPs, the performance of a given policy may depend on the number of trials/trajectories drawn to evaluate its performance (Mutti et al., 2023; Santos et al., 2024). In fact, the standard formulation of GUMDPs implicitly assumes the performance of a given policy is evaluated

over an infinite number of trials/trajectories of interaction with the environment. This is problematic because: (i) the infinite trials assumption is violated in many practical application domains where the objective function depends on the empirical occupancy induced by a small or finite set of trajectories; and (ii) in general, the optimal policies produced by algorithms from prior research may perform poorly when evaluated on a limited number of trajectories, as demonstrated by Mutti et al. (2023). To overcome this issue, previous research introduced a finite-trials formulation for GUMDPs where the objective function depends on the empirical occupancy induced by a finite set of trajectories (Mutti et al., 2023; Santos et al., 2024). Unfortunately, in the finite-horizon setting, Mutti et al. (2023) show that computing optimal policies for the finite-trials formulation of GUMDPs is computationally challenging, being significantly harder than its infinite trials counterpart. Specifically, the authors demonstrate that the problem can be reformulated as an "extended MDP" where the agent must keep track of the history of state-action pairs observed up to each timestep. Mutti et al. (2023) present preliminary results showing that optimal policies for the extended MDP, computed via dynamic programming techniques, outperform their infinite-trial counterparts. However, the state space of the extended MDP grows combinatorially with the horizon, limiting the scalability of the approach to very small problem instances.

In this work, we introduce the first approach for solving GUMDPs in the single-trial regime, i.e., when the agent's performance is evaluated based on a single trial/trajectory. We consider an infinite-horizon discounted setting, which has been greatly adopted by previous research in the field (Zahavy et al., 2021; Hazan et al., 2019) and has found important applications in different domains where the lifetime of the agent is uncertain or infinite. We focus our attention to environments with discrete state and action spaces. Our key contributions are threefold. First, we establish fundamental results on policy optimization in the single-trial regime, addressing: (i) which class of policies suffices for optimality; (ii) how the problem can be cast as an "occupancy MDP" that is equivalent to our original problem; and (iii) the computational complexity of policy optimization in the single-trial regime. Technically, our results differ from Mutti et al. (2023) due to the inherent differences between infinite-horizon discounted occupancies and the finite-horizon occupancies considered by the previous work. Moreover, our occupancy MDP refines the extended MDP from Mutti et al. (2023), preserves optimality guarantees, and is better suited for practical implementation. Second, we introduce a Monte-Carlo tree search (MCTS) algorithm to solve the occupancy MDP, effectively solving the GUMDP in the single-trial regime via online planning. Our approach provably retrieves the optimal action at each timestep for a sufficiently high number of iterations. Third, we present experimental results showcasing the superior performance of our approach over relevant baselines across diverse tasks and environments.

## 2 BACKGROUND

### 2.1 MARKOV DECISION PROCESSES

MDPs (Puterman, 2014) provide a mathematical framework to study sequential decision making and are formally defined as a tuple $\mathcal{M} = (\mathcal{S}, \mathcal{A}, \{\boldsymbol{P}^a : a \in \mathcal{A}\}, \boldsymbol{p}_0, c)$ where: $\mathcal{S}$ is the finite state space; $\mathcal{A}$ is the finite action space; $\{\boldsymbol{P}^a : a \in \mathcal{A}\}$ is a set of transition probability matrices $\boldsymbol{P}^a$, one for each action $a \in \mathcal{A}$; $\boldsymbol{p}_0 \in \Delta(\mathcal{S})$ is the initial state distribution; and $c : \mathcal{S} \times \mathcal{A} \to \mathbb{R}$ is the cost function. For a given action $a \in \mathcal{A}$, each row of matrix $\boldsymbol{P}^a$ satisfies $P^a(s, \cdot) \in \Delta(\mathcal{S})$, encoding the probability of transition from state $s$ at the present timestep to any other state at the next timestep when choosing action $a$. The interaction takes place as follows: (i) an initial state $\mathrm{s}_0$ is sampled from $\boldsymbol{p}_0$; (ii) at each step $t$, the agent observes the state of the environment $\mathrm{s}_t \in \mathcal{S}$ and chooses an action $\mathrm{a}_t \in \mathcal{A}$. Depending on the chosen action, the environment evolves to state $\mathrm{s}_{t+1} \in \mathcal{S}$ with probability $P^{\mathrm{a}_t}(\mathrm{s}_t, \cdot)$, and the agent receives a random cost $\mathrm{c}_t$ with expectation given by $c(\mathrm{s}_t, \mathrm{a}_t)$; and (iii) the interaction repeats infinitely.

A decision rule $\pi_t$ specifies the procedure for action selection at timestep $t$. A non-Markovian decision rule $\pi_t$, at each timestep $t$, maps the history of states and actions to a probability distribution over actions, i.e., $\pi_t : \mathcal{S} \times (\mathcal{S} \times \mathcal{A})^t \to \Delta(\mathcal{A})$. A Markovian decision rule does not take into account the entire history and, instead, maps the last state in the history to a distribution over actions, i.e., $\pi_t : \mathcal{S} \to \Delta(\mathcal{A})$. Both non-Markovian and Markovian decision rules can be deterministic if they consist of mappings of the type $\pi_t : \mathcal{S} \times (\mathcal{S} \times \mathcal{A})^t \to \mathcal{A}$ or $\pi_t : \mathcal{S} \to \mathcal{A}$, respectively.

A policy $\pi = (\pi_0, \pi_1, \ldots)$ is a sequence of decision rules, one for each timestep. If, for all timesteps, the decision rules are Markovian or non-Markovian, we say the policy is Markovian or non-Markovian, respectively. Similarly, if the decision rules are deterministic or stochastic for all timesteps, we say the policy is deterministic or stochastic, respectively. We denote the class of non-Markovian policies with $\Pi_{\mathrm{NM}}$, the class of Markovian policies with $\Pi_{\mathrm{M}}$, the class of non-Markovian deterministic policies with $\Pi_{\mathrm{NM}}^{\mathrm{D}}$, and the class of Markovian deterministic policies with $\Pi_{\mathrm{M}}^{\mathrm{D}}$. Finally, the class of stationary policies, $\Pi_{\mathrm{S}}$, contains all policies such that the decision rule is the same for all timesteps. We let $\Pi_{\mathrm{S}}^{\mathrm{D}}$ denote the class of stationary deterministic policies.

For a given policy $\pi \in \Pi_{\mathrm{NM}}$, the interaction between the agent and the environment is a random process $(s_0, a_0, s_1, a_1, \ldots)$. We let $h_t = (s_0, a_0, s_1, a_1, \ldots, s_t)$ denote a random history up to (including) timestep $t$. We also denote with $h_t = (s_0, a_0, s_1, a_1, \ldots, s_t) \in \mathcal{S} \times (\mathcal{S} \times \mathcal{A})^t$ a particular history up to timestep $t$. The random sequence $(s_0, a_0, s_1, a_1, \ldots)$ satisfies: (i) $\mathbb{P}[s_0 = s] = p_0(s)$; (ii) $\mathbb{P}[s_{t+1} = s'|h_t, a_t] = P^{a_t}(s_t, s')$; and (iii) $\mathbb{P}[a_t = a|h_t] = \pi_t(a|h_t)$. Let $(\Omega, \mathcal{F}, \mathbb{P}_\pi)$ be the probability space over the sequence of random variables $(s_0, a_0, s_1, a_1, \ldots)$ that satisfies conditions (i)-(iii) above (Lattimore & Szepesvári, 2020), where $\mathcal{F}$ is a sigma algebra. We write specific trajectories as $\omega \in \Omega$, with $\omega = (s_0, a_0, s_1, a_1, \ldots)$. We denote with $\mathbb{P}_\pi[s_t = s, a_t = a|s_0 \sim \boldsymbol{p}_0]$ the probability of state-action pair $(s, a)$ at timestep $t$ under policy $\pi$.

**The infinite-horizon discounted setting.** The discounted cumulative cost objective is $J_\gamma(\pi) = \mathbb{E}\left[\sum_{t=0}^\infty \gamma^t c(s_t, a_t)\right]$, where $\gamma \in (0, 1)$ is the discount factor and the expectation is taken over the random trajectory of state-action pairs $(s_0, a_0, s_1, a_1, \ldots)$ generated by the interaction between $\pi$ and the MDP. It is well-known that the class of stationary policies suffices for optimality (Puterman, 2014, Theo. 6.2.10) and, hence, we aim to find the optimal policy, $\pi^*$, such that $\pi^* = \arg\min_{\pi \in \Pi_{\mathrm{S}}} J_\gamma(\pi)$. The discounted state-action occupancy under policy $\pi$ is

$$d_\pi(s, a) = (1 - \gamma) \sum_{t=0}^\infty \gamma^t \mathbb{P}_\pi[s_t = s, a_t = a|s_0 \sim \boldsymbol{p}_0]. \tag{1}$$

The expected discounted cumulative cost of policy $\pi$ can be written as $J_\gamma(\pi) = \boldsymbol{c}^\top \boldsymbol{d}_\pi$, where $\boldsymbol{d}_\pi = [d_\pi(s_0, a_0), \ldots, d_\pi(s_{|\mathcal{S}|}, a_{|\mathcal{A}|})]^\top$ and $\boldsymbol{c} = [c(s_0, a_0), \ldots, c(s_{|\mathcal{S}|}, a_{|\mathcal{A}|})]^\top$. Then, the problem of computing the optimal policy becomes $\pi^* = \arg\min_{\pi \in \Pi_{\mathrm{S}}} \boldsymbol{c}^\top \boldsymbol{d}_\pi$, which can be formulated as a linear program (Puterman, 2014).

## 2.2 MONTE-CARLO TREE SEARCH

MCTS (Browne et al., 2012; Silver et al., 2017) is a sample-based planning algorithm to approximate optimal policies in MDPs through sequential tree-based search. The search tree alternates between decision nodes, representing agent actions, and chance nodes, representing stochastic environment transitions. At each iteration, MCTS builds and refines a search tree by alternating between four phases: selection, expansion, simulation, and backpropagation. In the selection phase, the algorithm recursively selects actions at decision nodes according to a tree policy, often based on upper confidence bounds, and samples successor states at chance nodes according to the environment's dynamics, until it reaches a node that has not yet been fully expanded. Then, in the expansion phase, a new child node corresponding to an unvisited state-action pair is created. In the simulation phase, a rollout policy (typically random or heuristic) generates a trajectory from the expanded node to estimate a Monte Carlo return. Backpropagation then updates the statistics (e.g., mean value, visit counts) along the path traversed during the selection phase. MCTS converges asymptotically to the optimal action at the root under mild assumptions (Kocsis & Szepesvári, 2006). Shah et al. (2020) and Cömer et al. (2025) establish polynomial regret concentration for MCTS-based algorithms.

## 2.3 GENERAL-UTILITY MARKOV DECISION PROCESSES

The GUMDP framework generalizes utility-specification by allowing the objective of the agent to be written in terms of the visitation frequency of state-action pairs. This is in contrast to the MDP framework, where the objective of the agent is encoded by the cost, a function of state-action pairs.

We define an infinite-horizon discounted GUMDP as a tuple $\mathcal{M}_f = (\mathcal{S}, \mathcal{A}, \{\boldsymbol{P}^a : a \in \mathcal{A}\}, \boldsymbol{p}_0, f)$ where $\mathcal{S}, \mathcal{A}, \{\boldsymbol{P}^a : a \in \mathcal{A}\}$, and $\boldsymbol{p}_0$ are defined in a similar way to the standard MDP formulation.

The objective of the agent is encoded by $f : \Delta(\mathcal{S} \times \mathcal{A}) \to \mathbb{R}$, as a function of a state-action discounted occupancy $\boldsymbol{d}$, as defined in equation 1. The objective is then to find

$$\pi^* = \underset{\pi \in \Pi_{\mathrm{S}}}{\arg\min} \, f(\boldsymbol{d}_\pi). \tag{2}$$

We highlight that, when $f$ is a linear function, we are under the standard MDP setting; if $f$ is convex, then we are under the convex MDP setting (Zahavy et al., 2021). In this work, we consider three different tasks, each associated with a particular (convex) objective function: (i) maximum state entropy exploration (Hazan et al., 2019), where $f(\boldsymbol{d}) = \boldsymbol{d}^\top \log(\boldsymbol{d})$; (ii) imitation learning (Abbeel & Ng, 2004), where $f(\boldsymbol{d}) = \|\boldsymbol{d} - \boldsymbol{d}_\beta\|_2^2$ and $\boldsymbol{d}_\beta \in \Delta(\mathcal{S} \times \mathcal{A})$ is the occupancy induced by behavior policy $\beta$; and (iii) adversarial MDPs (Rosenberg & Mansour, 2019), where $f(\boldsymbol{d}) = \max_{k \in \{1,\dots,K\}} \boldsymbol{d}^\top \boldsymbol{c}_k$ and $\{\boldsymbol{c}_1, \dots, \boldsymbol{c}_K\}$ is a set of $K$ cost vectors satisfying $c_k \in \mathbb{R}^{|\mathcal{S}||\mathcal{A}|}$. Nevertheless, our results apply to any task that can be modelled using the GUMDP framework. We refer to Zahavy et al. (2021) for a comprehensive list of the different objectives considered by previous works.

## 2.4 GUMDPs IN THE SINGLE-TRIAL REGIME

In this work, we consider a different objective from the one introduced in equation 2. While equation 2 depends on the *expected* discounted occupancy, $\boldsymbol{d}_\pi$, the objective we herein introduce depends on the *empirical* discounted occupancy induced by running a given policy on the GUMDP. This is particularly important, as practical applications often require identifying the policy that performs optimally when evaluated based on a single trajectory of interaction with the environment (Mutti et al., 2023; Santos et al., 2024). Furthermore, as we shall explain next, in GUMDPs the performance of a given policy may depend on the number of trajectories or trials used to evaluate it (Mutti et al., 2023; Santos et al., 2024).

**Discounted empirical state-action occupancies** We consider the setting in which the agent interacts with its environment over a single-trial, i.e., a single trajectory. For a given policy $\pi \in \Pi_{\mathrm{NM}}$, we introduce the random vector $\mathbf{d}^\pi : \Omega \to \Delta(\mathcal{S} \times \mathcal{A})$, which corresponds to the empirical discounted state-action occupancy associated with the probability space $(\Omega, \mathcal{F}, \mathbb{P}_\pi)$, defined as

$$\mathrm{d}_\omega^\pi(s, a) = (1 - \gamma) \sum_{t=0}^\infty \gamma^t \mathbf{1}(s_t = s, a_t = a), \tag{3}$$

where $\mathbf{1}$ is the indicator function. It holds that $d_\pi = \mathbb{E}[\mathbf{d}^\pi]$, for $d_\pi$ as introduced in equation 1. In practice, it is common to truncate the trajectories of interaction between the agent and its environment. We denote by $H \in \mathbb{N}$ the truncation horizon and let the empirical truncated occupancy, $\mathbf{d}^{\pi,H} : \Omega \to \Delta(\mathcal{S} \times \mathcal{A})$, be defined as

$$\mathrm{d}_\omega^{\pi,H}(s, a) = \frac{1 - \gamma}{1 - \gamma^H} \sum_{t=0}^{H-1} \gamma^t \mathbf{1}(s_t = s, a_t = a). \tag{4}$$

**Single-trial formulation for GUMDPs** We now introduce objectives for GUMDPs that depend on *empirical* discounted state-action occupancies. The *single-trial* objective is defined as

$$F_1(\pi) = \mathbb{E}\left[f(\mathbf{d}^\pi)\right], \tag{5}$$

and we aim to find $\pi^* = \arg\min_{\pi \in \Pi} F_1(\pi)$, where $\Pi$ is an arbitrary policy class we specify later. The *single-trial truncated* objective is defined as

$$F_{1,H}(\pi) = \mathbb{E}\left[f(\mathbf{d}^{\pi,H})\right]. \tag{6}$$

We note that the single-trial truncated objective is more general than the single-trial objective. In particular, $F_{1,H}$ is equivalent to $F_1$ as $H \to \infty$. The *infinite trials*[1] objective, $F_\infty$, is defined as

$$F_\infty(\pi) = f(\boldsymbol{d}_\pi) = f\left(\mathbb{E}\left[\mathbf{d}^\pi\right]\right),$$

and we aim to find $\pi_\infty^* = \arg\min_{\pi \in \Pi_{\mathrm{S}}} F_\infty(\pi)$. We note that $F_\infty$ is equivalent to the objective in equation 2, which depends on *expected* occupancies. The fact that $\Pi_{\mathrm{S}}$ suffices for optimality follows from results on the possible state-action occupancies induced by different classes of policies (Puterman, 2014).

---

[1] We call $F_\infty$ the *infinite trials* objective because, as the number of sampled trajectories/trials approaches infinity, the mismatch between GUMDPs that depend on empirical and expected occupancies fades away.

**The mismatch between $F_1$ and $F_\infty$** Previous works pointed out important differences between the single and infinite trials formulations for GUMDPs (Mutti et al., 2023; Santos et al., 2024). In particular, it has been shown that, in general, the performance of a given policy under the single and infinite trials formulations differs and, consequently, the optimal policy for each objective may also differ. This occurs because, since $f$ may be non-linear, it can happen that $F_1(\pi) = \mathbb{E}\left[f(\mathbf{d}^\pi)\right] \neq f\left(\mathbb{E}\left[\mathbf{d}^\pi\right]\right) = F_\infty(\pi)$. We refer to Santos et al. (2024) for explicit lower bounds on the performance difference between $F_1$ and $F_\infty$. Naturally, when $f$ is linear, as it is the case in standard MDPs, then the single and infinite trials formulations become equivalent due to the linearity of the expectation. However, due to the mismatch between the single and infinite trials formulations, and given that the single-trial formulation is particularly relevant in practical applications where policy performance is assessed based on a single trajectory of interaction with the environment, we focus in this work on finding (approximately) optimal policies for the single-trial objective, $F_1$.

## 3 POLICY OPTIMIZATION IN THE SINGLE-TRIAL REGIME

In this section, we establish the fundamental results that underpin the development of online planning algorithms to solve GUMDPs in the single-trial regime. Specifically, we investigate: (i) which class of policies suffices for optimality; (ii) how we can focus on the truncated single-trial objective, $F_{1,H}$, to compute approximately optimal policies for the single trial objective, $F_1$; (iii) how we can cast our single-trial GUMDP problem as an MDP in which the agent keeps track of the accrued occupancy at every timestep of the interaction with the GUMDP; and (iv) the computational complexity of policy optimization in the single-trial regime. We let $\mathrm{OptGap}(\pi) = F_1(\pi) - \min_{\pi' \in \Pi_{\mathrm{NM}}} F_1(\pi')$ be the optimality gap of an arbitrary policy $\pi \in \Pi_{\mathrm{NM}}$ with respect to the single-trial objective introduced in equation 5. Intuitively, the optimality gap measures how suboptimal a given policy $\pi$ is compared to the best policy. Throughout our work, we make use of the following assumption.

**Assumption 1.** *The objective function $f$ is $L$-Lipschitz with $L \in \mathbb{R}^+$, i.e., $|f(\mathbf{d}_1) - f(\mathbf{d}_2)| \leq L\|\mathbf{d}_1 - \mathbf{d}_2\|_1$ for any $\mathbf{d}_1, \mathbf{d}_2 \in \Delta(\mathcal{S} \times \mathcal{A})$.*

We refer to Appendix A for the Lipschitz constants of the objective functions considered.

### 3.1 NON-MARKOVIANITY MATTERS

We start by investigating which class of policies suffices for optimality. We have the following result (proof in Appendix B.1).

**Theorem 1.** *There exists a GUMDP $\mathcal{M}_f$ with $\gamma \in (0,1)$ and $L$-Lipschitz convex objective such that (lower is better):*

1. *$F_1(\pi_\mathrm{S}) > F_1(\pi_\mathrm{M})$, for some $\pi_\mathrm{M} \in \Pi_\mathrm{M}$ and any $\pi_\mathrm{S} \in \Pi_\mathrm{S}$.*

2. *$F_1(\pi_\mathrm{M}) > F_1(\pi_\mathrm{NM})$, for some $\pi_\mathrm{NM} \in \Pi_\mathrm{NM}$ and any $\pi_\mathrm{M} \in \Pi_\mathrm{M}$.*

The result above shows that, in general, the class of stationary policies is strictly dominated by the class of non-stationary policies, which is, in turn, strictly dominated by the class of non-Markovian policies. Hence, non-Markovianity matters, and we must focus our attention on history-dependent policies. Our Theo. 1 extends the result in Mutti et al. (2023), which considers finite-horizon GUMDPs, to the infinite-horizon discounted setting.

### 3.2 COMPUTING (APPROXIMATELY) OPTIMAL POLICIES BY RESORTING TO $F_{1,H}$

The result below (proof in Appendix B.2) establishes that the optimality gap $\mathrm{OptGap}(\pi)$ of any policy $\pi \in \Pi_{\mathrm{NM}}$ can be upper bounded, up to a constant, by the optimality gap of policy $\pi$ for the single-trial truncated objective.

**Proposition 1** (Optimality gap decomposition). *For arbitrary $\pi \in \Pi_{\mathrm{NM}}$, it holds that*

$$\mathrm{OptGap}(\pi) \leq \underbrace{F_{1,H}(\pi) - \min_{\pi_H \in \Pi_{\mathrm{NM}}} \{F_{1,H}(\pi_H)\}}_{= \, \mathrm{OptGap}_H(\pi)} + 8L\gamma^H, \tag{7}$$

*where $\mathrm{OptGap}_H(\pi)$ is the optimality gap of policy $\pi$ under the single-trial truncated objective with horizon $H$.*

Intuitively, the proposition above shows that we can resort to the single-trial truncated objective, as introduced in equation 6, to find an approximately optimal policy for the original objective defined in equation 5, up to any desired tolerance. In particular, if $\pi$ is the optimal policy for the single-trial truncated objective, i.e., $\mathrm{OptGap}_H(\pi) = 0$, then it holds that $\mathrm{OptGap}(\pi) \leq 8L\gamma^H$, which can be made arbitrarily small by tuning our truncation horizon $H$. Therefore, we focus our attention on finding

$$\pi^* = \underset{\pi \in \Pi_{\mathrm{NM}}}{\arg\min} \, F_{1,H}(\pi) = \underset{\pi \in \Pi_{\mathrm{NM}}}{\arg\min} \, \mathbb{E}\left[ f(\mathbf{d}^{\pi,H}) \right], \tag{8}$$

in order to keep the truncated optimality gap term $\mathrm{OptGap}_H(\pi)$ low, which we investigate in the next section.

## 3.3 THE OCCUPANCY MDP: CASTING $\mathcal{M}_f$ AS A STANDARD MDP

To derive our planning algorithms for solving GUMDPs in the single-trial truncated setting, we derive a finite-horizon MDP based on the original GUMDP. In particular, we consider the occupancy MDP defined by the tuple $\mathcal{M}_{\mathrm{O}} = \{\mathcal{S}_{\mathrm{O}}, \mathcal{A}_{\mathrm{O}}, \{\boldsymbol{P}_{\mathrm{O}}^a\}, \boldsymbol{p}_{0,\mathrm{O}}, c_{\mathrm{O}}, H\}$, where $\mathcal{S}_{\mathrm{O}} = \mathcal{S} \times \mathcal{O}$ is the discrete state space and

$$\mathcal{O} = \left\{ \boldsymbol{o} \in \mathbb{R}^{|\mathcal{S}||\mathcal{A}|} : o(s,a) = \sum_{t=0}^{l-1} \gamma^t \mathbf{1}(s_t = s, a_t = a), \forall s \in \mathcal{S}, a \in \mathcal{A}, \right.$$

$$\left. (s_0, a_0, \ldots, s_l) \in \mathcal{S} \times (\mathcal{S} \times \mathcal{A})^l, 1 \leq l \leq H-1 \right\} \bigcup \left\{ [0, \ldots, 0] \in \mathbb{R}^{|\mathcal{S}||\mathcal{A}|} \right\}.$$

We denote a state of the occupancy MDP with the tuple $\{s, \boldsymbol{o}\}$, where $s \in \mathcal{S}$ is a state from the original GUMDP and $\boldsymbol{o} \in \mathcal{O}$ is a $|\mathcal{S}||\mathcal{A}|$-dimensional vector that keeps track of the running occupancy of the agent up to a given timestep. Intuitively, the running occupancy records the empirical occupancy, as defined in equation 3, observed by the agent up to any timestep. We let $\mathcal{A}_{\mathrm{O}} = \mathcal{A}$ be the action space. We define $\boldsymbol{p}_{0,\mathrm{O}}$ such that $p_{0,\mathrm{O}}(\{s, \boldsymbol{o}\}) = p_0(s)$ if $\boldsymbol{o} = [0, \ldots, 0]$ and zero otherwise. The dynamics are as follows: (i) component $\mathrm{s}_{t+1} \sim P^{\mathrm{a}_t}(\cdot|\mathrm{s}_t)$ evolves according to the dynamics of the original GUMDP; and (ii) the running occupancy evolves deterministically as $o_{t+1}(s,a) = \gamma^t + o_t(s,a)$ if $s = \mathrm{s}_t$ and $a = \mathrm{a}_t$, and $o_{t+1}(s,a) = o_t(s,a)$ otherwise. We emphasize that we do not need to incorporate the timestep in the state of the occupancy MDP since it can be inferred from the running occupancy by summing its entries. Finally, $H \in \mathbb{N}$ denotes the horizon of the MDP and the cost function $c_{\mathrm{O}} : \mathcal{S} \times \mathcal{O} \to \mathbb{R}$ is defined as

$$c_{\mathrm{O}}(\{s, \boldsymbol{o}\}) = \begin{cases} 0 & \text{if } t < H, \\ f\left( \frac{1-\gamma}{1-\gamma^H} \boldsymbol{o} \right) & \text{if } t = H. \end{cases}$$

Stationary policies $\pi_{\mathrm{O}} \in \Pi_{\mathrm{S}}$ for $\mathcal{M}_{\mathrm{O}}$ are mappings of the type $\pi_{\mathrm{O}} : \mathcal{S} \times \mathcal{O} \to \Delta(\mathcal{A})$. We let the cumulative cost under $\mathcal{M}_{\mathrm{O}}$ be

$$J_{\mathrm{O}}(\pi_{\mathrm{O}}) = \mathbb{E}\left[ \sum_{t=0}^{H} c_{\mathrm{O}}(\{s_t, \mathbf{o}_t\}) \right] = \mathbb{E}\left[ c_{\mathrm{O}}(\{s_H, \mathbf{o}_H\}) \right], \tag{9}$$

where the expectation above is taken with respect to the random sequence of states $(\{s_0, \mathbf{o}_0\}, \ldots, \{s_H, \mathbf{o}_H\})$ under policy $\pi_{\mathrm{O}}$. We let $J_{\mathrm{O}}^* = \min_{\pi_{\mathrm{O}} \in \Pi_{\mathrm{S}}^{\mathrm{o}}} J_{\mathrm{O}}(\pi_{\mathrm{O}})$ be the optimal cumulative cost for $\mathcal{M}_{\mathrm{O}}$. We also note that the occupancy MDP possesses well-defined (optimal) value and action-value functions, which can be shown to satisfy standard Bellman equations (Appendix B.3).

We present the following result, relating states in $\mathcal{M}_{\mathrm{O}}$ to histories in $\mathcal{M}_f$ (proof in Appendix B.4).

**Proposition 2** (One-to-one mapping between histories in $\mathcal{M}_f$ and states in $\mathcal{M}_{\mathrm{O}}$). *There exists a one-to-one mapping between histories* $h_l = (s_0, a_0, s_1, a_1, \ldots, s_l) \in \mathcal{S} \times (\mathcal{S} \times \mathcal{A})^l$ *in* $\mathcal{M}_f$, *with* $0 \leq l \leq H-1$, *and states* $\{s, \boldsymbol{o}\} \in \mathcal{S} \times \mathcal{O}$ *in* $\mathcal{M}_{\mathrm{O}}$.

An important conclusion that follows from the result above is that there exists a one-to-one mapping between non-Markovian policies for $\mathcal{M}_f$ and stationary policies for $\mathcal{M}_{\mathrm{O}}$. This holds because every state in $\mathcal{M}_{\mathrm{O}}$ is uniquely associated with a particular history in $\mathcal{M}_f$ (and vice versa). With this in mind, we now state the following result (proof in Appendix B.5), which connects the problem of solving the occupancy MDP and the problem of solving the single-trial truncated GUMDP objective.

**Theorem 2** (Solving $\mathcal{M}_f$ is "equivalent" to solving $\mathcal{M}_O$). *The problem of finding a policy $\pi \in \Pi_{\text{NM}}$ satisfying $\text{OptGap}_H(\pi) \leq \epsilon$, for any $\epsilon \in \mathbb{R}_0^+$, can be reduced to the problem of finding a policy $\pi_O \in \Pi_S$ satisfying $J_O(\pi_O) - J_O^* \leq \epsilon$. In particular, if $\pi_O^* = \arg\min_{\pi_o \in \Pi_S} J_O(\pi_O)$, then the corresponding non-Markovian policy $\pi$ in $\mathcal{M}_f$ satisfies $\text{OptGap}_H(\pi) = 0$. Finally, it holds that $\text{OptGap}_H(\pi) = J_O(\pi_O) - J_O^*$, where $\pi_O$ is the stationary policy for $\mathcal{M}_O$ associated with the non-Markovian policy $\pi$ for $\mathcal{M}_f$.*

Intuitively, the result above tells us that it suffices to search for an approximately stationary optimal policy for $\mathcal{M}_O$, since such a policy corresponds to a non-Markovian policy that is approximately optimal for $\mathcal{M}_f$. In particular, an approximately optimal policy for $\mathcal{M}_O$ can be seen as a non-Markovian policy for $\mathcal{M}_f$ that compresses the history up to any timestep into a running occupancy. This result demonstrates that maintaining the running occupancy up to any timestep is sufficient to achieve optimal behavior in the single-trial truncated-horizon regime equation 8.

**Remark 1** (Deterministic policies suffice for optimality). *Since the class of policies $\Pi_S^D$ suffices for optimality in standard MDPs (Puterman, 2014), we know that at least one stationary deterministic policy $\pi_O \in \Pi_S^D$ for $\mathcal{M}_O$ satisfies $J_O(\pi_O) - J_O^* = 0$, i.e., $\pi_O$ is optimal for $\mathcal{M}_O$. In light of Theo. 2 and Prop. 2, this implies that the corresponding non-Markovian policy $\pi \in \Pi_{\text{NM}}$ for $\mathcal{M}_f$, which is deterministic, satisfies $\text{OptGap}_H(\pi) = 0$, i.e., $\pi$ is optimal for $\mathcal{M}_f$. Thus, we can focus our attention on deterministic non-Markovian policies when solving $\mathcal{M}_f$ with objective $F_{1,H}$, i.e., for any GUMDP and horizon $H \in \mathbb{N}$, it holds that $\min_{\pi \in \Pi_{\text{NM}}} F_{1,H}(\pi) = \min_{\pi \in \Pi_{\text{NM}}^D} F_{1,H}(\pi)$.*

Given Theo. 2, we consider planning algorithms to solve the occupancy MDP. Unfortunately, solving the occupancy MDP poses some challenges: (i) the cost function of the occupancy MDP is rather sparse since it is only non-zero at the last timestep; (ii) the size of the state space of the occupancy MDP grows combinatorially with $H$ since every state in the occupancy MDP is associated with a possible history in $\mathcal{M}_f$; and (iii) every state in the occupancy MDP is visited at most once per trajectory. Therefore, before investigating how we can solve the occupancy MDP in Sec. 4, we take a closer look at how hard it is, from a worst-case perspective, to compute the optimal policy in GUMDPs in the single-trial regime.

It is worth noting that the occupancy MDP is conceptually related to the extended MDP proposed by Mutti et al. (2023) for the case of undiscounted finite-horizon GUMDPs. While the extended MDP explicitly tracks the full history up to the current timestep, we show that this information can be compressed into a running occupancy without sacrificing optimality guarantees. Since there exists a one-to-one mapping between states of the occupancy MDP and histories, the size of the state space of both formulations is equivalent. However, the compressed representation used by the occupancy MDP is more amenable to practical implementations since the running occupancy can be incrementally updated as the agent interacts with its environment. Despite these similarities, a key distinction lies in the setting: Mutti et al. (2023) consider the finite-horizon undiscounted setting, whereas we focus on the discounted setting. Discounting plays a crucial role in our analysis (e.g., Proposition 2) and it remains unclear whether similar results hold in the undiscounted case. This highlights a fundamental difference between our work and that of Mutti et al. (2023).

### 3.4 HARDNESS RESULT FOR POLICY OPTIMIZATION IN THE SINGLE-TRIAL REGIME

In the previous section, we established that it suffices to search over the class of policies $\Pi_{\text{NM}}^D$ in order to attain optimal policies for any GUMDP and horizon $H \in \mathbb{N}$ with respect to objective $F_{1,H}(\pi) = \mathbb{E}\left[f(\mathbf{d}^{\pi,H})\right]$. We now show that there exist GUMDPs for which solving $\pi^* = \arg\min_{\pi \in \Pi_{\text{NM}}^D} F_{1,H}(\pi)$ can be computationally hard. More precisely, we prove that the problem of deciding whether there exists a policy $\pi \in \Pi_{\text{NM}}^D$ such that $F_{1,H}(\pi) \leq \lambda$, where $\lambda \in \mathbb{R}$ is a threshold value, is NP-Hard.

**Theorem 3** (NP-Hardness of policy optimization in the single-trial regime). *Given a GUMDP with objective $F_{1,H}$ and a threshold value $\lambda \in \mathbb{R}$, it is NP-Hard to determine whether there exists a policy $\pi \in \Pi_{\text{NM}}^D$ satisfying $F_{1,H}(\pi) \leq \lambda$.*

*Proof sketch.* (Complete proof in Appendix B.6) We reduce the subset sum problem to the policy existence problem in GUMDPs with objective $F_{1,H}$. The subset sum problem asks whether, given a set $\mathcal{N} = \{n_0, n_1, \ldots, n_{N-1}\}$ of $N$ non-negative integers and a target sum $k \in \mathbb{N}$, there exists

a subset of $\mathcal{N}$ whose elements sum to $k$. We map every instance of the subset sum problem as a GUMDP such that at each state $s_i$, for $i \in \{0, \ldots, N - 1\}$, the policy $\pi \in \Pi_{\text{NM}}^{\text{D}}$ needs to decide between selecting: (i) $a_{\text{include}}$, thereby including $n_i$ in the sum; or (ii) $a_{\text{not-include}}$, thereby excluding $n_i$ from the sum. Then, we set $H \geq N$ and let $f(\boldsymbol{d}) = (\boldsymbol{n}^\top \boldsymbol{d} - k)^2$, where $\boldsymbol{d}$ denotes a discounted occupancy that captures information regarding the actions selected by the agent at each state $s_i$. We construct vector $\boldsymbol{n} \in \mathbb{R}^{|\mathcal{S}||\mathcal{A}|}$ such that $\boldsymbol{n}^\top \boldsymbol{d}$ equals the sum of the numbers selected by the policy. With this construction, the objective satisfies $f(\boldsymbol{d}) = 0$ if and only if the sum of the selected numbers equals $k$. By setting $\lambda = 0$, we are asking whether there exists a policy such that $F_{1,H}(\pi) \leq 0$. Since $f(\boldsymbol{d}) = 0$ if and only if the selected numbers sum to $k$, the reduction is complete. □

We note that the objective function $f$ used in the proof of the result above is Lipschitz and (strictly) convex. Thus, our result shows that, even for smooth convex objectives, the computational hardness of computing the optimal policy in the single-trial regime is NP-Hard. Mutti et al. (2023) present a hardness result for the single-trial optimization problem in the case of undiscounted finite-horizon GUMDPs. Our theorem extends this result to the discounted case. In addition, our proof is significantly simpler - a one-step reduction - compared to the NP-hardness argument in Mutti et al. (2023), which relies on complexity results for partially observable MDPs. Furthermore, our result is more informative, as it shows that the hardness persists even when the objective $f$ is smooth and convex.

With the above hardness result in mind, we next explore how to develop practical planning algorithms for our problem. Naturally, in a worst-case sense, these algorithms may require a non-polynomial number of steps to retrieve the optimal policy. Nevertheless, our results show it is possible to develop practical algorithms that are superior in comparison to relevant baselines.

## 4 ONLINE PLANNING FOR GUMDPS IN THE SINGLE-TRIAL REGIME

In this section, we investigate how we can solve the occupancy MDP introduced in the previous section by resorting to online planning techniques.

As previously shown, solving a GUMDP in the single-trial setting is closely related to solving a corresponding occupancy MDP. This connection allows us to employ an online planning approach in which, at any timestep $t \in \{0, \ldots, H - 1\}$ of the interaction with the occupancy MDP, the algorithm receives the current state $\{s_t, \boldsymbol{o}_t\}$. The online planner then expands a look-ahead search tree where the root node corresponds to state $\{s_t, \boldsymbol{o}_t\}$. After a given number of iterations, the planning algorithm selects an action to execute in the environment; depending on the selected action, the environment evolves to a new state, and the process repeats until timestep $H$. This online planning strategy is particularly effective, as it allows computational resources to be focused on computing (approximately) optimal actions only along the specific trajectory experienced by the agent. This avoids the prohibitive cost of computing an optimal policy for every state of the occupancy MDP.

### 4.1 MONTE-CARLO TREE SEARCH FOR GUMDPS IN THE SINGLE-TRIAL REGIME

We employ an MCTS algorithm to solve the occupancy MDP. As described in Sec. 2.2, the search tree of the online planning algorithm comprises decision and chance nodes. In the context of the occupancy MDP, each decision node corresponds to an action $a \in \mathcal{A}$, while each chance node corresponds to a given state $\{s, \boldsymbol{o}\} \in \mathcal{S} \times \mathcal{O}$ of the occupancy MDP. At timestep $t$ of the interaction, the MCTS algorithm builds a planning tree rooted at the current state $\{s_t, \boldsymbol{o}_t\}$, following the four phases outlined in Sec. 2.2 at each iteration. We provide the following two remarks, which further characterize the convergence and regret of our proposed MCTS-based approach for solving GUMDPs in the single-trial regime. To derive the two results below, we make the following assumption.

**Assumption 2.** *The objective function satisfies $f_{min} \leq f(\boldsymbol{d}) \leq f_{max}$, for any $\boldsymbol{d} \in \Delta(\mathcal{S} \times \mathcal{A})$.*

**Remark 2** (Convergence). *Under Assumption 2, for any horizon $H \in \mathbb{N}$, the MCTS algorithm provably solves the occupancy MDP as the number of iterations of the algorithm per timestep grows to infinity. More precisely, for any horizon $H \in \mathbb{N}$, we have that $\mathrm{OptGap}_H(\pi_{MCTS}) = 0$ as the number of iterations of the algorithm per timestep grows to infinity, where we let $\pi_{MCTS}$ be the policy induced by the MCTS algorithm at each timestep. This result follows from our Theo. 2 and Theo. 6 in Kocsis & Szepesvári (2006) by rescaling the objective function to lie in the $[0, 1]$ interval. Thus, from Prop. 1, $\mathrm{OptGap}(\pi_{MCTS}) \leq 8 \frac{L}{f_{max} - f_{min}} \gamma^H$.*

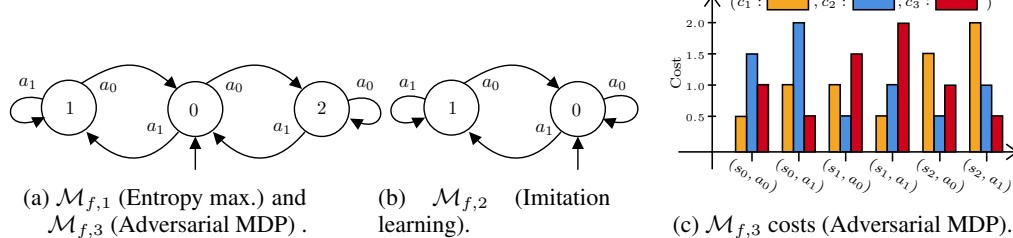

(a) $\mathcal{M}_{f,1}$ (Entropy max.) and $\mathcal{M}_{f,3}$ (Adversarial MDP) .

(b) $\mathcal{M}_{f,2}$ (Imitation learning).

(c) $\mathcal{M}_{f,3}$ costs (Adversarial MDP).

Figure 1: Illustrative GUMDPs. $\mathcal{M}_{f,1}$ and $\mathcal{M}_{f,3}$ share the same dynamics but differ in the objective function. In all GUMDPs, the chosen action succeeds with $90\%$ probability and, with $10\%$ probability, the agent randomly moves to any of the states. The behavior policy for $\mathcal{M}_{f,2}$ is $\beta(a_0|s_0) = 0.8$ and $\beta(a_0|s_1) = 0.2$. In (c), we plot the three cost functions, $c_1, c_2$ and $c_3$, of the adversarial MDP.

**Remark 3** (Regret bound). *For a given number of episodes $n$ and a sequence of policies $(\pi_1, \ldots, \pi_n)$, let $\mathcal{R}(\pi_1, \ldots, \pi_n) = \frac{1}{n}\sum_{i=1}^{n} OptGap(\pi_i)$ be the expected regret for the sequence of policies and $\mathcal{R}_H(\pi_1, \ldots, \pi_n) = \frac{1}{n}\sum_{i=1}^{n} OptGap_H(\pi_i)$ the truncated expected regret for the sequence of policies. Under Assumption 2, for any horizon $H$, there exists an MCTS-based algorithm such that, when applied to solve the occupancy MDP with $n$ expansion steps, yields a sequence of policies $(\pi_1, \ldots, \pi_n)$ satisfying $\mathcal{R}_H(\pi_1, \ldots, \pi_n) \leq O(1/\sqrt{n})$. Furthermore, from Prop. 1, $\mathcal{R}(\pi_1, \ldots, \pi_n) \leq O(1/\sqrt{n} + \gamma^H)$. This result follows from our Theo. 2 and the works of Shah et al. (2020) and Cömer et al. (2025), particularly Theo. 1 in Shah et al. (2020) and Theo. 1 in Cömer et al. (2025), which establish polynomial regret bounds for MCTS-based algorithms.*

## 5  EXPERIMENTAL RESULTS

In this section, we empirically assess the performance of the proposed MCTS-based algorithm for solving GUMDPs in the single-trial setting. Below, we provide a brief description of the considered tasks and environments. We refer to Appendix C for a complete description of our experiments.

**Tasks, environments, baselines, and experimental methodology**  We consider three tasks: (i) maximum state entropy exploration (Hazan et al., 2019); imitation learning (Abbeel & Ng, 2004); and (iii) adversarial MDPs (Rosenberg & Mansour, 2019). We consider two sets of environments. The first set consists of the illustrative GUMDPs depicted in Fig. 1, each associated with one of the tasks. The second set of environments come from the OpenAI Gym library (Brockman et al., 2016). We consider the FrozenLake (FL), Taxi, and mountaincar (MC) environments. The framework of GUMDPs is defined over discrete state spaces; hence, we discretized the MC environment using a $10 \times 10$ grid with equally-spaced bins. For the FL, Taxi, and MC environments, the task of imitation learning consists in imitating an approximately optimal policy. We let $\gamma = 0.9$ and set $H = 100$ for the illustrative GUMDPs and $H = 200$ for the other environments. We perform 10 runs per experimental setting. We consider two baselines: (i) a random policy, $\pi_{\text{Random}}$; and (ii) the optimal policy for the infinite trials formulation equation 2, $\pi_{\text{Solver}}^*$, calculated by solving a constrained optimization problem with objective $f$ via a standard optimization solver. We denote the policy induced by our MCTS algorithm as $\pi_{\text{MCTS}}$ and consider 4000 iterations per timestep (results with other numbers of iterations in the Appendix C). Our code can be found here

**Experimental results discussion**  We present our experimental results in Tab. 1. As seen, across nearly all experimental settings, $\pi_{\text{MCTS}}$ outperformed the baselines, showcasing the superior performance of our approach (the only exception is for the Taxi environment under the imitation learning task where the performance of $\pi_{\text{MCTS}}$ is similar to that of $\pi_{\text{Solver}}^*$). We highlight the gains attained by $\pi_{\text{MCTS}}$ in comparison to the infinite trials policy, $\pi_{\text{Solver}}^*$.

## 6  CONCLUSION & LIMITATIONS

In this work, we contribute with the first approach to solve infinite-horizon discounted GUMDPs in the single-trial regime. In Sec. 3, we provided the fundamental results underpinning policy optimiza-

Table 1: Mean single-trial objective, $F_{1,H}(\pi)$, obtained by different policies, across tasks and environments. Values in parentheses correspond to the $90\%$ mean conf. interval. Lower is better.

| Policy | Maximum state entropy exploration | | | | Imitation learning | | | | Adversarial MDP |
|---|---|---|---|---|---|---|---|---|---|
| | $\mathcal{M}_{f,1}$ | FL | Taxi | MC | $\mathcal{M}_{f,2}$ | FL | Taxi | MC | $\mathcal{M}_{f,3}$ |
| $\pi_{\text{Random}}$ | 0.12 (-0.04,+0.04) | 0.51 (-0.03,+0.03) | 0.65 (-0.01,+0.01) | 0.72 (-0.02,+0.02) | 0.05 (-0.02,+0.02) | 0.07 (-0.02,+0.02) | 0.08 (-0.01,+0.01) | 0.18 (-0.04,+0.04) | 1.23 (-0.02,+0.02) |
| $\pi^*_{\text{Solver}}$ | 0.05 (-0.02,+0.02) | 0.48 (-0.03,+0.03) | 0.63 (-0.01,+0.01) | 0.70 (-0.03,+0.03) | 0.02 (-0.01,+0.01) | 0.05 (-0.02,+0.02) | **0.05** (-0.001,+0.002) | 0.07 (-0.01,+0.02) | 1.17 (-0.02,+0.02) |
| $\pi_{\text{MCTS}}$ | **0.01** (-0.01,+0.01) | **0.40** (-0.03,+0.03) | **0.59** (-0.0,+0.0) | **0.61** (-0.02,+0.01) | **0.002** (-0.001,+0.001) | **0.02** (-0.005,+0.006) | **0.05** (-0.002,+0.002) | **0.04** (-0.01,+0.01) | **1.07** (-0.003,+0.004) |

tion in the discounted single-trial regime. Then, in Secs. 4 and 5, we explored how we can resort to online planning techniques, in particular MCTS, to solve discounted GUMDPs in the single-trial regime. Our work takes a first step towards a broader application of GUMDPs in real-world settings where the agent's performance is typically evaluated under a single trial. The key limitations of our approach to solve GUMDPs in the single-trial regime are: (i) the MCTS algorithm requires a simulator of the environment to sample transitions; and (ii) the size of the matrix that keeps track of the running occupancy may be impractical for GUMDPs with large state and action spaces. We believe such limitations should be addressed by future work, for example, by investigating methods to compress the running occupancy.

### ACKNOWLEDGMENTS

This work was supported by Portuguese national funds through the Portuguese Fundação para a Ciência e a Tecnologia (FCT) under projects UID/50021/2025 and UID/PRR/50021/2025 (INESC-ID multi-annual funding), as well as AI-PackBot (project number 14935, LISBOA2030-FEDER-00854700). Pedro P. Santos acknowledges the FCT PhD grant 2021.04684.BD. Alberto Sardinha acknowledges the CNPq Research Productivity Fellowship (PQ), with reference 312699/2025-5. The authors also thank the lab managers at GAIPS for the support provided when running the computational experiments of this work.

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

## A  LIPSCHITZ CONSTANTS

Table 2: Common objective functions found in the GUMDPs literature. In (†) we assume $\boldsymbol{d}$ is lower bounded by $\epsilon$ satisfying $0 < \epsilon < e^{-2}$.

| Task | Objective ($f(\boldsymbol{d})$) | Lipschitz constant ($L$) |
|:---:|:---:|:---:|
| MDPs/RL | $\boldsymbol{d}^\top \boldsymbol{c}, \quad \boldsymbol{c} \in \mathbb{R}^{|\mathcal{S}||\mathcal{A}|}$ | $\max_{s,a} |c(s,a)|$ |
| Pure exploration | $\boldsymbol{d}^\top \log(\boldsymbol{d})$ | $|\log(\epsilon) + 1|$ (†) |
| Imitation learning | $\|\boldsymbol{d} - \boldsymbol{d}_\beta\|_2^2, \quad \boldsymbol{d}_\beta \in \Delta(\mathcal{S} \times \mathcal{A})$ | 4 |
| Adversarial MDPs | $\max_{k \in \{1,\dots,K\}} \boldsymbol{d}^\top \boldsymbol{c}_k$ | $\max_{k \in \{1,\dots,K\}} \{\max_{s,a} |c_k(s,a)|\}$ |

**Objective function $f(\boldsymbol{d}) = \boldsymbol{c}^\top \boldsymbol{d}$**  It holds that

$$|f(\boldsymbol{d}_1) - f(\boldsymbol{d}_2)| = |\boldsymbol{c}^\top (\boldsymbol{d}_1 - \boldsymbol{d}_2)| = \sum_{s,a} |c(s,a)||d_1(s,a) - d_2(s,a)| \leq \max_{s,a} |c(s,a)| \|\boldsymbol{d}_1 - \boldsymbol{d}_2\|_1.$$

**Objective function $f(\boldsymbol{d}) = \boldsymbol{d}^\top \log(\boldsymbol{d})$**  We assume $\boldsymbol{d}$ is lower bounded by $\epsilon$, i.e., $d(s,a) \geq \epsilon$ with $0 < \epsilon < e^{-2}$ for all $s \in \mathcal{S}, a \in \mathcal{A}$. We let $f(\boldsymbol{d}) = \sum_{s,a} g(d(s,a))$, for $g(x) = x \log(x)$. We note that, $g'(x) = \log(x) + 1$ and it holds for any $x \in [\epsilon, 1]$ that $|g'(x)| \leq |\log(\epsilon) + 1|$. Thus, for any $x_1, x_2 \in [\epsilon, 1]$ we have that

$$\begin{aligned}
|g(x_1) - g(x_2)| = \left| \int_{x_2}^{x_1} g'(x) dx \right| &= \left| \int_{\min\{x_1,x_2\}}^{\max\{x_1,x_2\}} g'(x) dx \right| \\
&\leq \int_{\min\{x_1,x_2\}}^{\max\{x_1,x_2\}} |g'(x)| \, dx \leq \int_{\min\{x_1,x_2\}}^{\max\{x_1,x_2\}} |\log(\epsilon) + 1| \, dx \\
&= |\log(\epsilon) + 1| \, |x_1 - x_2|.
\end{aligned}$$

Thus, for any $\boldsymbol{d}_1, \boldsymbol{d}_2 \in \Delta(\mathcal{S} \times \mathcal{A})$ lower bounded by $0 < \epsilon < e^{-2}$, it holds that

$$\begin{aligned}
|f(\boldsymbol{d}_1) - f(\boldsymbol{d}_1)| &= \left| \sum_{s,a} (g(d_1(s,a)) - g(d_2(s,a))) \right| \\
&\overset{(a)}{\leq} \sum_{s,a} |g(d_1(s,a)) - g(d_2(s,a))| \\
&\leq \sum_{s,a} |\log(\epsilon) + 1| \, |d_1(s,a) - d_2(s,a)| \\
&= |\log(\epsilon) + 1| \, \|\boldsymbol{d}_1 - \boldsymbol{d}_2\|_1
\end{aligned}$$

were (a) follows from the triangular inequality.

**Objective function $f(\boldsymbol{d}) = \|\boldsymbol{d} - \boldsymbol{d}_\beta\|_2^2$**  It holds that $\nabla f(\boldsymbol{d}) = 2(\boldsymbol{d} - \boldsymbol{d}_\beta)$. Now,

$$\max_{\boldsymbol{d} \in \Delta(\mathcal{S} \times \mathcal{A})} \|\nabla f(\boldsymbol{d})\|_1 = 2 \max_{\boldsymbol{d} \in \Delta(\mathcal{S} \times \mathcal{A})} \|\boldsymbol{d} - \boldsymbol{d}_\beta\|_1 \leq 2 \max_{\boldsymbol{d}_1, \boldsymbol{d}_2 \in \Delta(\mathcal{S} \times \mathcal{A})} \|\boldsymbol{d}_1 - \boldsymbol{d}_2\|_1 = 4.$$

Since the function $f$ is continuous and differentiable over the simplex, which is compact, it holds that $L = 4$ is a valid Lipschitz constant as it corresponds to an upper bound on the maximum magnitude of the gradient of $f$ over $\Delta(\mathcal{S} \times \mathcal{A})$.

## B  SUPPLEMENTARY MATERIALS FOR SEC. 3

### B.1  PROOF OF THEOREM 1

**Theorem 1.** *There exists a GUMDP $\mathcal{M}_f$ with $\gamma \in (0,1)$ and $L$-Lipschitz convex objective such that:*

1. $F_1(\pi_S) > F_1(\pi_M)$, *for some* $\pi_M \in \Pi_M$ *and any* $\pi_S \in \Pi_S$.

2. $F_1(\pi_M) > F_1(\pi_{NM})$, *for some* $\pi_{NM} \in \Pi_{NM}$ *and any* $\pi_M \in \Pi_M$.

*Proof.* To prove our result, we consider the GUMDP depicted in Fig. 2. To simplify our proof, we consider state-dependent occupancies and denote an occupancy for the GUMDP above with the vector $\boldsymbol{d} = [d(s^0), d(s^1), d(s^2)]$. The objective function is $f(\boldsymbol{d}) = \boldsymbol{d}^\top \boldsymbol{A}\boldsymbol{d}$, where $A = \mathrm{diag}([0, 1, 1])$, which is Lipschitz (over $\Delta(\mathcal{S})$) and convex. Under any trajectory it holds that $d(s^0) = (1-\gamma)\sum_{t=0}^\infty \gamma^{2t+1} = \frac{(1-\gamma)\gamma}{1-\gamma^2}$. Hence, it holds that, under any trajectory, $d(s^1) + d(s^2) = 1 - \frac{(1-\gamma)\gamma}{1-\gamma^2} = \frac{1-\gamma}{1-\gamma^2}$. Thus, we focus our attention to the value of the occupancy at state $s^1$, $d(s^1)$, and let $d(s^2) = \frac{1-\gamma}{1-\gamma^2} - d(s^1)$. With this, we can define our objective as a function of $d(s^1)$ only by letting $f(d(s^1)) = d(s^1)^2 + (\frac{1-\gamma}{1-\gamma^2} - d(s^1))^2$.

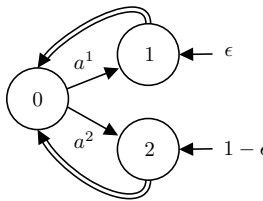

Figure 2: Illustration of the GUMDP used in the proof of Theo. 1 with $\mathcal{S} = \{s^0, s^1, s^2\}$ and $\mathcal{A} = \{a^1, a^2\}$. The distribution of initial states is $p_0(s^0) = 0, p_0(s^1) = \epsilon, p_0(s^2) = 1 - \epsilon$, where we set $\epsilon = 1/2$. All transitions are deterministic and in states $s^1$ and $s^2$ any of the actions takes the agent back to state $s^0$.

For the first part of the Theorem it holds, for any $\pi_S \in \Pi_S$, that

$$F_1(\pi_S) = \mathbb{E}\left[f(\mathbf{d}^{\pi_S})\right]$$

$$\overset{(a)}{=} \mathbb{E}\left[f(\mathrm{d}^{\pi_S}_{s_0,a_0,s_1,a_1,\ldots}(s^1)) \,\middle|\, \begin{smallmatrix} s_0 \sim p_0, a_0 \sim \pi_S(\cdot|s_0), \\ s_1 \sim P^{a_0}(\cdot|s_0), a_1 \sim \pi_S(\cdot|s_1), \\ \ldots, \\ s_3 \sim P^{a_2}(\cdot|s_2), a_3 \sim \pi_S(\cdot|s_3), \ldots \end{smallmatrix}\right]$$

$$\overset{(b)}{=} \mathbb{E}\left[\mathbb{E}\left[f(\mathrm{d}^{\pi_S}_{s_0,a_0,s_1,a_1,\ldots}(s^1)) \,\middle|\, \begin{smallmatrix} a_5 \sim \pi_S(\cdot|s^0), \\ s_6 \sim P^{a_5}(\cdot|s^0), \ldots \end{smallmatrix}\right] \,\middle|\, \begin{smallmatrix} s_0 \sim p_0, a_0 \sim \pi_S(\cdot|s_0), \\ s_1 \sim P^{a_0}(\cdot|s_0), a_1 \sim \pi_S(\cdot|s_1), \\ \ldots, \\ s_4 \sim P^{a_3}(\cdot|s_3), a_4 \sim \pi_S(\cdot|s_4) \end{smallmatrix}\right]$$

$$\overset{(c)}{=} \mathbb{E}\left[\mathbb{E}\left[f\left((1-\gamma)\sum_{t=0}^4 \gamma^t \mathbf{1}(s_t = s^1) + \gamma^5 \tilde{\mathrm{d}}^{\pi_S}_{a_5,s_6,a_6,\ldots}(s^1)\right) \,\middle|\, \begin{smallmatrix} a_5 \sim \pi_S(\cdot|s^0), \\ s_6 \sim P^{a_5}(\cdot|s^0), \ldots \end{smallmatrix}\right] \,\middle|\, \begin{smallmatrix} s_0 \sim p_0, a_0 \sim \pi_S(\cdot|s_0), \\ s_1 \sim P^{a_0}(\cdot|s_0), a_1 \sim \pi_S(\cdot|s_1), \\ \ldots, \\ s_4 \sim P^{a_3}(\cdot|s_3), a_4 \sim \pi_S(\cdot|s_4) \end{smallmatrix}\right]$$

$$\overset{(d)}{\geq} \mathbb{E}\left[f\left((1-\gamma)\sum_{t=0}^4 \gamma^t \mathbf{1}(s_t = s^1) + \gamma^5 \mathbb{E}\left[\tilde{\mathrm{d}}^{\pi_S}_{a_5,s_6,a_6,\ldots}(s^1) \,\middle|\, \begin{smallmatrix} a_5 \sim \pi_S(\cdot|s^0), \\ s_6 \sim P^{a_5}(\cdot|s^0), \ldots \end{smallmatrix}\right]\right) \,\middle|\, \begin{smallmatrix} s_0 \sim p_0, a_0 \sim \pi_S(\cdot|s_0), \\ s_1 \sim P^{a_0}(\cdot|s_0), a_1 \sim \pi_S(\cdot|s_1), \\ \ldots, \\ s_4 \sim P^{a_3}(\cdot|s_3), a_4 \sim \pi_S(\cdot|s_4) \end{smallmatrix}\right]$$

$$= \mathbb{E}\left[f\left((1-\gamma)\left(\mathbf{1}(s_0 = s^1) + \gamma^2 \mathbf{1}(s_2 = s^1) + \gamma^4 \mathbf{1}(s_4 = s^1)\right)\right.\right.$$

$$\left.\left. + (1-\gamma)\mathbb{E}\left[\sum_{t=5}^\infty \gamma^t \mathbf{1}(s_t = s^1) \,\middle|\, \begin{smallmatrix} a_5 \sim \pi_S(\cdot|s^0), \\ s_6 \sim P^{a_5}(\cdot|s^0), \ldots \end{smallmatrix}\right]\right) \,\middle|\, \begin{smallmatrix} s_0 \sim p_0, a_0 \sim \pi_S(\cdot|s_0), \\ s_1 \sim P^{a_0}(\cdot|s_0), a_1 \sim \pi_S(\cdot|s_1), \\ \ldots, \\ s_4 \sim P^{a_3}(\cdot|s_3), a_4 \sim \pi_S(\cdot|s_4) \end{smallmatrix}\right]$$

$$\overset{(e)}{=} \mathbb{E}\left[f\left((1-\gamma)\left(\mathbf{1}(s_0 = s^1) + \gamma^2 \mathbf{1}(s_2 = s^1) + \gamma^4 \mathbf{1}(s_4 = s^1)\right)\right.\right.$$

$$\left.\left. + (1-\gamma)\pi_S(a^1|s^0)\frac{\gamma^6}{1-\gamma^2}\right) \,\middle|\, \begin{smallmatrix} s_0 \sim p_0, a_0 \sim \pi_S(\cdot|s_0), \\ s_1 \sim P^{a_0}(\cdot|s_0), a_1 \sim \pi_S(\cdot|s_1), \\ \ldots, \\ s_4 \sim P^{a_3}(\cdot|s_3), a_4 \sim \pi_S(\cdot|s_4) \end{smallmatrix}\right],$$

where in (a) we emphasized that the random vector $\mathbf{d}^{\pi_S}$ depends on random variables $s_0, a_0, s_1, a_1, \ldots$; (b) follows from the fact that $s_5 = s^0$ with probability one for trajectories drawn from the GUMDP, i.e., for any trajectory the state at timestep 5 is always $s^0$, and thus we can

split and simplify the expectation. In step (c), we let $\tilde{\mathbf{d}}$ be the random vector defined as in equation 3 for the GUMDP depicted in Fig. 2, but where $p_0(s^0) = 1$ and zero otherwise. Step (d) follows from Jensen's inequality since, for any values $(s_0, a_0, s_1, a_1, \ldots, s_4, a_4) \in (\mathcal{S} \times \mathcal{A})^5$ the random variables of the outer expectation can take, it holds that $\mathbb{E}[g(\tilde{\mathbf{d}}^{\pi_S})] \geq g(\mathbb{E}[\tilde{\mathbf{d}}^{\pi_S}])$ where we let $g(x) = f((1-\gamma)\sum_{t=0}^{4}\gamma^t \mathbf{1}(s_t = s^1) + \gamma^5 x)$, which is convex. Finally, step (e) follows from the fact that

$$\mathbb{E}\left[\sum_{t=5}^{\infty}\gamma^t \mathbf{1}(\mathbf{s}_t = s^1)\Bigg|_{\mathbf{s}_6 \sim P^{a_5}(\cdot|s^0), \ldots}^{a_5 \sim \pi_S(\cdot|s^0),}\right] = \sum_{t=5}^{\infty}\gamma^t \mathbb{P}_{\pi_S}\left[\mathbf{s}_t = s^1\right]$$

$$= \gamma^5 \cdot 0 + \gamma^6 \pi_S(a^1|s^0) + \gamma^7 \cdot 0 + \gamma^8 \pi_S(a^1|s^0) + \ldots$$

$$= \pi_S(a^1|s^0)\left(\gamma^6 + \gamma^8 + \ldots\right)$$

$$= \pi_S(a^1|s^0)\sum_{t=0}^{\infty}\gamma^{2t+6}$$

$$= \pi_S(a^1|s^0)\frac{\gamma^6}{1-\gamma^2}.$$

We can now explicitly write the expectation in the last step above, yielding, for any $\pi_S \in \Pi_S$ and while letting $\epsilon = 1/2$,

$$F_1(\pi_S) \geq \frac{1}{2}\pi_S(a^1|s^0)^2\left[f\left((1-\gamma)\left(1+\gamma^2+\gamma^4+\pi_S(a^1|s^0)\frac{\gamma^6}{1-\gamma^2}\right)\right)\right.$$

$$\left. + f\left((1-\gamma)\left(\gamma^2+\gamma^4+\pi_S(a^1|s^0)\frac{\gamma^6}{1-\gamma^2}\right)\right)\right]$$

$$+ \frac{1}{2}\pi_S(a^1|s^0)(1-\pi_S(a^1|s^0))\left[f\left((1-\gamma)\left(1+\gamma^2+\pi_S(a^1|s^0)\frac{\gamma^6}{1-\gamma^2}\right)\right)\right.$$

$$+ f\left((1-\gamma)\left(1+\gamma^4+\pi_S(a^1|s^0)\frac{\gamma^6}{1-\gamma^2}\right)\right)$$

$$+ f\left((1-\gamma)\left(\gamma^2+\pi_S(a^1|s^0)\frac{\gamma^6}{1-\gamma^2}\right)\right)$$

$$\left. + f\left((1-\gamma)\left(\gamma^4+\pi_S(a^1|s^0)\frac{\gamma^6}{1-\gamma^2}\right)\right)\right]$$

$$+ \frac{1}{2}(1-\pi_S(a^1|s^0))^2\left[f\left((1-\gamma)\left(1+\pi_S(a^1|s^0)\frac{\gamma^6}{1-\gamma^2}\right)\right)\right.$$

$$\left. + f\left((1-\gamma)\left(\pi_S(a^1|s^0)\frac{\gamma^6}{1-\gamma^2}\right)\right)\right].$$

In summary, $F_1(\pi_S)$ is lower bounded by the expression above for any policy $\pi_S \in \Pi_S$. Since $f$ is a quadratic function, the lower bound above is also a quadratic function with respect to variable $\pi_S(a^1|s^0)$. Thus, we can calculate the minimizer of the lower bound above by computing the gradient with respect to $\pi_S(a^1|s^0)$ and setting it to zero. It can be checked that, for any $\gamma \in (0,1)$, $\pi_S(a^1|s^0) = 1/2$ minimizes the lower bound above (we provide below a snippet of Mathematica code that supports this claim). This implies that, for any $\pi_S \in \Pi_S$, $F_1(\pi_S)$ is lower bounded by the expression above evaluated at $\pi_S(a^1|s^0) = 1/2$, i.e.,

$$F_1(\pi_S) \geq \frac{2 - 2\gamma^2 + 2\gamma^4 - 2\gamma^6 + 2\gamma^8 - 2\gamma^{10} + \gamma^{12}}{2(1+\gamma)^2}.$$

Now, consider the non-stationary policy $\pi_M \in \Pi_M$ that deterministically selects $a^1$ at timesteps $t = 3, 7, 11, \ldots$ and deterministically selects $a^2$ at timesteps $t = 1, 5, 9, \ldots$. It holds that

$$F_1(\pi_M) = \mathbb{E}\left[f(\mathbf{d}^{\pi_M})\right]$$

$$= \mathbb{E}\left[f\left((1-\gamma)\sum_{t=0}^{\infty}\gamma^t \mathbf{1}(\mathrm{s}_t = s^1)\right)\right]$$

$$= \epsilon f\left((1-\gamma)(1+0+0+0+\gamma^4+0+0+0+\gamma^8+\ldots)\right)$$
$$+ (1-\epsilon)f\left((1-\gamma)(0+0+0+0+\gamma^4+0+0+0+\gamma^8+\ldots)\right)$$

$$= \epsilon f\left((1-\gamma)\left(1+\sum_{t=0}^{\infty}\gamma^{4t+4}\right)\right) + (1-\epsilon)f\left((1-\gamma)\sum_{t=0}^{\infty}\gamma^{4t+4}\right)$$

$$\overset{(a)}{=} \frac{1}{2}f\left((1-\gamma)\left(1+\frac{\gamma^4}{1-\gamma^4}\right)\right) + \frac{1}{2}f\left(\frac{(1-\gamma)\gamma^4}{1-\gamma^4}\right)$$

$$= \frac{1+\gamma^2-\gamma^6+\gamma^8}{(1-\gamma)^2(1+\gamma^2)^2},$$

where in (a) we let $\epsilon = 1/2$ and simplified the sums.

To conclude, it holds, for any $\pi_S \in \Pi_S$ and $\gamma \in (0,1)$, that

$$F_1(\pi_S) \geq \frac{2-2\gamma^2+2\gamma^4-2\gamma^6+2\gamma^8-2\gamma^{10}+\gamma^{12}}{2(1+\gamma)^2} > \frac{1+\gamma^2-\gamma^6+\gamma^8}{(1-\gamma)^2(1+\gamma^2)^2} = F_1(\pi_M),$$

which can be verified using a software for symbolic/algebraic computation such as Mathematica (Wolfram Research). We provide a snippet of the Mathematica code we used below.

Snippet of Mathematica code to support the proof that $F_1(\pi_S) > F_1(\pi_M), \forall \pi_S$.

```
[In]: f[o_, g_] := o^2 + (((1 - g)/(1 - g^2)) - o)^2
[In]: h[x_, g_] := (1/2)*x^2*(f[(1 - g) (1 + g^2 + g^4 + x*(g^6/(1 - g^2))), g] +
        f[(1 - g) (g^2 + g^4 + x*(g^6/(1 - g^2))), g]) +
    (1/2)*x (1 - x)*(f[(1 - g) (1 + g^2 + x*(g^6/(1 - g^2))), g] +
        f[(1 - g) (1 + g^4 + x*(g^6/(1 - g^2))), g] +
        f[(1 - g) (g^2 + x*(g^6/(1 - g^2))), g] +
        f[(1 - g) (g^4 + x*(g^6/(1 - g^2))), g]) +
    (1/2)*(1 - x)^2 (f[(1 - g) (1 + x*(g^6/(1 - g^2))), g] +
        f[(1 - g) (x*(g^6/(1 - g^2))), g])
[In]: Simplify[Solve[D[h[x, g], x] == 0, x]]
[Out]: {{x -> 1/2}}
[In]: a[g_] = (1/2)*f[(1 - g)*(1 + g^4/(1 - g^4)), g] + (1/2)*f[(1 - g)*(g^4/(1 - g^4)), g]
[In]: Reduce[a[g] < h[1/2, g]]
[Out]: g < 0 || 0 < g < 1 || g > 1
```

For the second part of the Theorem it holds, for any $\pi_M = (\pi_0, \pi_1, \pi_2, \ldots) \in \Pi_M$, that

$$F_1(\pi_M) = \mathbb{E}\left[f(\mathbf{d}^{\pi_M})\right]$$

$$\overset{(a)}{=} \mathbb{E}\left[f(\mathrm{d}^{\pi_M}_{\mathrm{s}_0,\mathrm{a}_0,\mathrm{s}_1,\mathrm{a}_1,\ldots}(s^1))\Big|_{\substack{\mathrm{s}_0\sim p_0,\mathrm{a}_0\sim\pi_0(\cdot|\mathrm{s}_0),\\ \mathrm{s}_1\sim P^{\mathrm{a}_0}(\cdot|\mathrm{s}_0),\mathrm{a}_1\sim\pi_1(\cdot|\mathrm{s}_1),\\ \mathrm{s}_2\sim P^{\mathrm{a}_1}(\cdot|\mathrm{s}_1),\mathrm{a}_2\sim\pi_2(\cdot|\mathrm{s}_2),\ldots}}\right]$$

$$\overset{(b)}{=} \mathbb{E}\left[\mathbb{E}\left[f(\mathrm{d}^{\pi_M}_{\mathrm{s}_0,\mathrm{a}_0,\mathrm{s}_1,\mathrm{a}_1,\ldots}(s^1))\Big|_{\substack{\mathrm{a}_3\sim\pi_3(\cdot|s^0),\\ \mathrm{s}_4\sim P^{\mathrm{a}_3}(\cdot|s^0),\ldots}}\right]\Big|_{\substack{\mathrm{s}_0\sim p_0,\mathrm{a}_0\sim\pi_0(\cdot|\mathrm{s}_0),\\ \mathrm{s}_1\sim P^{\mathrm{a}_0}(\cdot|\mathrm{s}_0),\mathrm{a}_1\sim\pi_1(\cdot|\mathrm{s}_1),\\ \mathrm{s}_2\sim P^{\mathrm{a}_1}(\cdot|\mathrm{s}_1),\mathrm{a}_2\sim\pi_2(\cdot|\mathrm{s}_2)}}\right]$$

$$\overset{(c)}{\geq} \mathbb{E}\left[f\left(\mathbb{E}\left[\mathrm{d}^{\pi_M}_{\mathrm{s}_0,\mathrm{a}_0,\mathrm{s}_1,\mathrm{a}_1,\ldots}(s^1)\Big|_{\substack{\mathrm{a}_3\sim\pi_3(\cdot|s^0),\\ \mathrm{s}_4\sim P^{\mathrm{a}_3}(\cdot|s^0),\ldots}}\right]\right)\Big|_{\substack{\mathrm{s}_0\sim p_0,\mathrm{a}_0\sim\pi_0(\cdot|\mathrm{s}_0),\\ \mathrm{s}_1\sim P^{\mathrm{a}_0}(\cdot|\mathrm{s}_0),\mathrm{a}_1\sim\pi_1(\cdot|\mathrm{s}_1),\\ \mathrm{s}_2\sim P^{\mathrm{a}_1}(\cdot|\mathrm{s}_1),\mathrm{a}_2\sim\pi_2(\cdot|\mathrm{s}_2)}}\right]$$

$$= \mathbb{E}\left[f\left(\mathbb{E}\left[(1-\gamma)\sum_{t=0}^{\infty}\gamma^t \mathbf{1}(\mathrm{s}_t = s^1)\Big|_{\substack{\mathrm{a}_3\sim\pi_3(\cdot|s^0),\\ \mathrm{s}_4\sim P^{\mathrm{a}_3}(\cdot|s^0),\ldots}}\right]\right)\Big|_{\substack{\mathrm{s}_0\sim p_0,\mathrm{a}_0\sim\pi_0(\cdot|\mathrm{s}_0),\\ \mathrm{s}_1\sim P^{\mathrm{a}_0}(\cdot|\mathrm{s}_0),\mathrm{a}_1\sim\pi_1(\cdot|\mathrm{s}_1),\\ \mathrm{s}_2\sim P^{\mathrm{a}_1}(\cdot|\mathrm{s}_1),\mathrm{a}_2\sim\pi_2(\cdot|\mathrm{s}_2)}}\right]$$

$$= \mathbb{E}\left[f\left((1-\gamma)\left(\mathbf{1}(\mathrm{s}_0 = s^1) + \mathbf{1}(\mathrm{s}_2 = s^1)\right.\right.\right.$$
$$\left.\left.\left. + \mathbb{E}\left[\sum_{t=3}^{\infty}\gamma^t \mathbf{1}(\mathrm{s}_t = s^1)\Big|_{\substack{\mathrm{a}_3\sim\pi_3(\cdot|s^0),\\ \mathrm{s}_4\sim P^{\mathrm{a}_3}(\cdot|s^0),\ldots}}\right]\right)\right)\Big|_{\substack{\mathrm{s}_0\sim p_0,\mathrm{a}_0\sim\pi_0(\cdot|\mathrm{s}_0),\\ \mathrm{s}_1\sim P^{\mathrm{a}_0}(\cdot|\mathrm{s}_0),\mathrm{a}_1\sim\pi_1(\cdot|\mathrm{s}_1),\\ \mathrm{s}_2\sim P^{\mathrm{a}_1}(\cdot|\mathrm{s}_1),\mathrm{a}_2\sim\pi_2(\cdot|\mathrm{s}_2)}}\right],$$

where in (a) we emphasized that the random vector $\mathbf{d}^{\pi_M}$ depends on random variables $\mathrm{s}_0, \mathrm{a}_0, \mathrm{s}_1, \mathrm{a}_1, \ldots$. Step (b) follows from the fact that $\mathrm{s}_3 = s^0$ with probability one for trajectories drawn from the GUMDP, i.e., for any trajectory the state at timestep 3 is always $s^0$, and thus we

can split and simplify the expectation. Step (c) follows from Jensen's inequality, following similar steps as those for the first part of the Theorem. Now, it holds that

$$\mathbb{E}\left[\sum_{t=3}^{\infty}\gamma^t \mathbf{1}(s_t = s^1)\Big|\begin{smallmatrix}a_3 \sim \pi_3(\cdot|s^0),\\ s_4 \sim P^{a_3}(\cdot|s^0),\dots\end{smallmatrix}\right] = \sum_{t=3}^{\infty}\gamma^t \mathbb{P}_{\pi_M}\left[s_t = s^1\right]$$

$$= \gamma^3 \cdot 0 + \gamma^4 \pi_3(a^1|s^0) + \gamma^5 \cdot 0 + \gamma^6 \pi_5(a^1|s^0) + \dots$$

$$= \sum_{t=2}^{\infty}\gamma^{2t}\pi_{2t-1}(a^1|s^0).$$

For any policy $\pi_M$, it holds that $\sum_{t=2}^{\infty}\gamma^{2t}\pi_{2t-1}(a^1|s^0) \in [0, \frac{\gamma^4}{1-\gamma^2}]$. Hence, if we replace expression $\mathbb{E}\left[\sum_{t=3}^{\infty}\gamma^t \mathbf{1}(s_t = s^1)\Big|\begin{smallmatrix}a_3 \sim \pi_3(\cdot|s^0),\\ s_4 \sim P^{a_3}(\cdot|s^0),\dots\end{smallmatrix}\right]$ with $c\frac{\gamma^4}{1-\gamma^2}$, for $c \in [0,1]$, and show that

$$\mathbb{E}\left[f\left((1-\gamma)\left(\mathbf{1}(s_0 = s^1) + \mathbf{1}(s_2 = s^1) + c\frac{\gamma^4}{1-\gamma^2}\right)\right)\Big|\begin{smallmatrix}s_0 \sim p_0, a_0 \sim \pi_0(\cdot|s_0),\\ s_1 \sim P^{a_0}(\cdot|s_0), a_1 \sim \pi_1(\cdot|s_1),\\ s_2 \sim P^{a_1}(\cdot|s_1), a_2 \sim \pi_2(\cdot|s_2)\end{smallmatrix}\right]$$

is strictly lower bounded by $F_1(\pi_{NM})$ for a given $\pi_{NM} \in \Pi_{NM}$, for any $\pi_0, \pi_1, \pi_2 \in \Pi_S$ and $c \in [0,1]$, this implies that $F_1(\pi_{NM})$ is strictly lower than that of any possible $\pi \in \Pi_M$. For any $\pi_0, \pi_1, \pi_2 \in \Pi_S$ and $c \in [0,1]$, the expectation in the expression above can be simplified as

$$\mathbb{E}\left[f\left((1-\gamma)\left(\mathbf{1}(s_0 = s^1) + \mathbf{1}(s_2 = s^1) + c\frac{\gamma^4}{1-\gamma^2}\right)\right)\Big|\begin{smallmatrix}s_0 \sim p_0, a_0 \sim \pi_0(\cdot|s_0),\\ s_1 \sim P^{a_0}(\cdot|s_0), a_1 \sim \pi_1(\cdot|s_1),\\ s_2 \sim P^{a_1}(\cdot|s_1), a_2 \sim \pi_2(\cdot|s_2)\end{smallmatrix}\right]$$

$$= \epsilon\pi_1(a^1|s^0)f\left((1-\gamma)\left(1+\gamma^2 + c\frac{\gamma^4}{1-\gamma^2}\right)\right)$$

$$+ \epsilon(1-\pi_1(a^1|s^0))f\left((1-\gamma)\left(1+c\frac{\gamma^4}{1-\gamma^2}\right)\right)$$

$$+ (1-\epsilon)\pi_1(a^1|s^0)f\left((1-\gamma)\left(\gamma^2 + c\frac{\gamma^4}{1-\gamma^2}\right)\right)$$

$$+ (1-\epsilon)(1-\pi_1(a^1|s^0))f\left((1-\gamma)\left(c\frac{\gamma^4}{1-\gamma^2}\right)\right)$$

$$\overset{(a)}{=} \frac{1}{2}\pi_1(a^1|s^0)\left(f\left((1-\gamma)\left(1+\gamma^2 + c\frac{\gamma^4}{1-\gamma^2}\right)\right) + f\left((1-\gamma)\left(\gamma^2 + c\frac{\gamma^4}{1-\gamma^2}\right)\right)\right)$$

$$+ \frac{1}{2}(1-\pi_1(a^1|s^0))\left(f\left((1-\gamma)\left(1+c\frac{\gamma^4}{1-\gamma^2}\right)\right) + f\left((1-\gamma)\left(c\frac{\gamma^4}{1-\gamma^2}\right)\right)\right),$$

where in (a) we let $\epsilon = 1/2$.

Now consider the non-markovian policy $\pi_{NM} \in \Pi_{NM}$ that: (i) if $s_0 = s^1$, then at timesteps $t = 1, 5, 9, \dots$ deterministically selects action $a^2$ and at timesteps $t = 3, 7, 11, \dots$ deterministically action $a^1$; (ii) if $s_0 = s^2$, then at timesteps $t = 1, 5, 9, \dots$ deterministically selects action $a^1$ and at timesteps $t = 3, 7, 11, \dots$ deterministically action $a^2$. We have that

$$F_1(\pi_{NM}) = \mathbb{E}\left[f(\mathbf{d}^{\pi_{NM}})\right]$$

$$= \mathbb{E}\left[f\left((1-\gamma)\sum_{t=0}^{\infty}\gamma^t \mathbf{1}(s_t = s^1)\right)\right]$$

$$= \epsilon f\left((1-\gamma)(1+0+0+0+\gamma^4+0+0+0+\gamma^8+\dots)\right)$$

$$+ (1-\epsilon)f\left((1-\gamma)(0+0+\gamma^2+0+0+0+\gamma^6+0+0+0+\gamma^{10}+\dots)\right)$$

$$= \epsilon f\left((1-\gamma)\sum_{t=0}^{\infty}\gamma^{4t}\right) + (1-\epsilon)f\left((1-\gamma)\sum_{t=0}^{\infty}\gamma^{4t+2}\right)$$

$$\overset{(a)}{=} \frac{1}{2}f\left(\frac{1-\gamma}{1-\gamma^4}\right) + \frac{1}{2}f\left(\frac{(1-\gamma)\gamma^2}{1-\gamma^4}\right)$$

where in (a) we let $\epsilon = 1/2$ and simplified the sums.

We now need to verify that, for any $\pi_1(a^1|s^0) \in [0, 1]$, $c \in [0, 1]$ and $\gamma \in (0, 1)$,

$$\pi_1(a^1|s^0) \underbrace{\frac{1}{2} \left( f\left( (1-\gamma) \left( 1 + \gamma^2 + c\frac{\gamma^4}{1-\gamma^2} \right) \right) + f\left( (1-\gamma) \left( \gamma^2 + c\frac{\gamma^4}{1-\gamma^2} \right) \right) \right)}_{(i)}$$

$$+ (1 - \pi_1(a^1|s^0)) \underbrace{\frac{1}{2} \left( f\left( (1-\gamma) \left( 1 + c\frac{\gamma^4}{1-\gamma^2} \right) \right) + f\left( (1-\gamma) \left( c\frac{\gamma^4}{1-\gamma^2} \right) \right) \right)}_{(ii)}$$

$$> \frac{1}{2} f\left( \frac{1-\gamma}{1-\gamma^4} \right) + \frac{1}{2} f\left( \frac{(1-\gamma)\gamma^2}{1-\gamma^4} \right) = F_1(\pi_{\mathrm{NM}}).$$

As can be seen, the expression on the left-hand side of the inequality above corresponds to a weighted combination (with weights $\pi_1(a^1|s^0)$ and $1-\pi_1(a^1|s^0)$) of components (i) and (ii). By resorting to a software for symbolic/algebraic computation such as Mathematica (Wolfram Research) it can be shown that $(i) > \frac{1}{2} f\left( \frac{1-\gamma}{1-\gamma^4} \right) + \frac{1}{2} f\left( \frac{(1-\gamma)\gamma^2}{1-\gamma^4} \right) = F_1(\pi_{\mathrm{NM}})$ and $(ii) > \frac{1}{2} f\left( \frac{1-\gamma}{1-\gamma^4} \right) + \frac{1}{2} f\left( \frac{(1-\gamma)\gamma^2}{1-\gamma^4} \right) = F_1(\pi_{\mathrm{NM}})$ for any $c \in [0, 1]$ and $\gamma \in (0, 1)$. We provide the snippets of the Mathematica code we used below. This implies that the weighted combination satisfies the inequality above for any $\pi_1(a^1|s^0) \in [0, 1]$ and the conclusion follows.

Snippet of Mathematica code to attest that $(i) > F_1(\pi_{\mathrm{NM}})$.

```
[In]: f[o_, g_] := o^2 + (((1 - g)/(1 - g^2)) - o)^2
[In]: m[c_, g_] := (1/2)*(f[(1 - g)*(1 + g^2 + c*(g^4/(1 - g^2))), g] +
    f[(1 - g)*(g^2 + c*(g^4/(1 - g^2))), g])
[In]: n[g_] := (1/2)*f[(1 - g)/(1 - g^4), g] + (1/2)*f[((1 - g)*g^2)/(1 - g^4), g]
[In]: Reduce[m[c, g] > n[g]]
[Out]: (c < 1/2 && (g < -1 || -1 < g < 0 || g > 0)) ||
    (c == 1/2 && (g < -1 || -1 < g < 0 || 0 < g < 1 || g > 1)) ||
    (c > 1/2 && (g < -1 || -1 < g < 0 || g > 0))
```

Snippet of Mathematica code to attest that $(ii) > F_1(\pi_{\mathrm{NM}})$.

```
[In]: f[o_, g_] := o^2 + (((1 - g)/(1 - g^2)) - o)^2
[In]: m[c_, g_] := (1/2)*(f[(1 - g)*(1 + c*(g^4/(1 - g^2))), g] +
    f[(1 - g)*(c*(g^4/(1 - g^2))), g])
[In]: n[g_] := (1/2)*f[(1 - g)/(1 - g^4), g] + (1/2)*f[((1 - g)*g^2)/(1 - g^4), g]
[In]: Reduce[m[c, g] > n[g]]
[Out]: (c < 1/2 && (g < -1 || -1 < g < 0 || g > 0)) ||
    (c == 1/2 && (g < -1 || -1 < g < 0 || 0 < g < 1 || g > 1)) ||
    (c > 1/2 && (g < -1 || -1 < g < 0 || g > 0))
```

$\square$

## B.2 PROOF OF PROPOSITION 1

**Lemma 1.** *For any $\omega \in \Omega$, $\pi \in \Pi_{\mathrm{NM}}$ and $H \in \mathbb{N}$ it holds that $\left| f(\mathbf{d}_\omega^\pi) - f(\mathbf{d}_\omega^{\pi,H}) \right| \leq 2L\gamma^H$.*

*Proof.* For any $\omega \in \Omega$, $\pi \in \Pi_{\mathrm{NM}}$ and $H \in \mathbb{N}$ it holds that

$$\left| f(\mathbf{d}_\omega^\pi) - f(\mathbf{d}_\omega^{\pi,H}) \right| \overset{(a)}{\leq} L \left\| \mathbf{d}_\omega^\pi - \mathbf{d}_\omega^{\pi,H} \right\|_1$$

$$\overset{(b)}{=} L \left\| (1-\gamma) \sum_{t=0}^{\infty} \gamma^t \mathbf{d}_{\omega,t}^\pi - \frac{(1-\gamma)}{1-\gamma^H} \sum_{t=0}^{H-1} \gamma^t \mathbf{d}_{\omega,t}^\pi \right\|_1$$

$$= L \left\| \frac{(1-\gamma)}{1-\gamma^H} \sum_{t=0}^{H-1} \gamma^t \left( (1-\gamma^H)\mathbf{d}_{\omega,t}^\pi - \mathbf{d}_{\omega,t}^\pi \right) + (1-\gamma) \sum_{t=H}^{\infty} \gamma^t \mathbf{d}_{\omega,t}^\pi \right\|_1$$

$$\overset{(c)}{\leq} L \left( \frac{(1-\gamma)}{1-\gamma^H} \sum_{t=0}^{H-1} \gamma^t \left\| (1-\gamma^H)\mathbf{d}_{\omega,t}^\pi - \mathbf{d}_{\omega,t}^\pi \right\|_1 + (1-\gamma) \sum_{t=H}^{\infty} \gamma^t \|\mathbf{d}_{\omega,t}^\pi\|_1 \right)$$

$$= L\frac{(1-\gamma)}{1-\gamma^H}\gamma^H \sum_{t=0}^{H-1}\gamma^t \left\|\mathbf{d}_{\omega,t}^\pi\right\|_1 + L\gamma^H$$
$$= 2L\gamma^H,$$

where: (a) is due to the $L$-Lipschitz assumption; in (b) we used $\mathbf{d}_\omega^\pi = (1-\gamma)\sum_{t=0}^\infty \gamma^t \mathbf{d}_{\omega,t}^\pi$ where $\mathbf{d}_{\omega,t}^\pi(s,a) = \mathbf{1}(s_t = s, a_t = a)$ is the empirical occupancy induced by the trajectory $\omega$ at timestep $t$ and $\mathbf{d}_\omega^{\pi,H} = (1-\gamma)/(1-\gamma^H)\sum_{t=0}^{H-1}\gamma^t \mathbf{d}_{\omega,t}^\pi$. Step (c) follows from the triangular inequality. $\square$

**Lemma 2.** *If $f$ is $L$-Lipschitz then it holds, for arbitrary $\pi \in \Pi_{\mathrm{NM}}$ and $H \in \mathbb{N}$, that*

$$|F_1(\pi) - F_{1,H}(\pi)| \le 2L\gamma^H.$$

*Proof.* It holds that, for arbitrary $\pi \in \Pi_{\mathrm{NM}}$ and $H \in \mathbb{N}$,

$$\begin{aligned}
|F_1(\pi) - F_{1,H}(\pi)| &= \left|\mathbb{E}\left[f(\mathbf{d}^\pi)\right] - \mathbb{E}\left[f(\mathbf{d}^{\pi,H})\right]\right| \\
&= \left|\mathbb{E}\left[f(\mathbf{d}^\pi) - f(\mathbf{d}^{\pi,H})\right]\right| \\
&\stackrel{(a)}{\le} \mathbb{E}\left[\left|f(\mathbf{d}^\pi) - f(\mathbf{d}^{\pi,H})\right|\right] \\
&= \sum_{\omega \in \Omega}\mathbb{P}_\pi\left[\omega\right]\left|f(\mathbf{d}_\omega^\pi) - f(\mathbf{d}_\omega^{\pi,H})\right| \\
&\stackrel{(b)}{\le} 2L\gamma^H,
\end{aligned}$$

where: (a) follows from $|\mathbb{E}[X]| \le \mathbb{E}[|X|]$; and (b) is due to Lemma 1. $\square$

**Lemma 3.** *For every GUMDP $\mathcal{M}_f$ with $L$-Lipschitz $f$ and $H \in \mathbb{N}$, if $\pi^* = \arg\min_{\pi \in \Pi_{\mathrm{NM}}} F_{1,H}(\pi)$, then it holds that $\mathrm{OptGap}(\pi^*) \le 4L\gamma^H$.*

*Proof.* As shown in Lemma 2, $|F_1(\pi) - F_{1,H}(\pi)| \le 2L\gamma^H$, for arbitrary $\pi$. From such inequality, we can infer that $F_{1,H}(\pi) - 2L\gamma^H \le F_1(\pi), \forall \pi \in \Pi_{\mathrm{NM}}$, i.e., function $F_{1,H}(\pi) - 2L\gamma^H$ lower bounds function $F_1(\pi)$. We provide a visual illustration of $F_1$ and $F_{1,H}$ in Fig. 3. Let $\pi^* = \arg\min_\pi F_{1,H}(\pi)$. It holds that

$$F_{1,H}(\pi^*) - 2L\gamma^H = \min_\pi F_{1,H}(\pi) - 2L\gamma^H \stackrel{(a)}{\le} \min_\pi F_1(\pi) \stackrel{(b)}{\le} F_1(\pi^*),$$

where (a) follows from the fact that $F_{1,H}(\pi) - 2L\gamma^H$ lower bounds $F_1(\pi)$; and (b) from the fact that $\min_\pi F_1(\pi) \le F_1(\pi'), \forall \pi'$ (from the definition of a minimum). We illustrate the inequalities above in Fig. 3. Finally, we note that

$$\begin{aligned}
F_1(\pi^*) - \left(F_{1,H}(\pi^*) - 2L\gamma^H\right) &= F_1(\pi^*) - F_{1,H}(\pi^*) + 2L\gamma^H \\
&\le |F_1(\pi^*) - F_{1,H}(\pi^*)| + 2L\gamma^H \\
&\le 4L\gamma^H.
\end{aligned}$$

The above implies that

$$\mathrm{OptGap}(\pi^*) = F_1(\pi^*) - \min_\pi F_1(\pi) \le 4L\gamma^H,$$

as illustrated in Fig. 3.

$\square$

**Proposition 1** (Optimality gap decomposition). *For arbitrary $\pi \in \Pi_{\mathrm{NM}}$, it holds that*

$$\mathrm{OptGap}(\pi) \le \underbrace{F_{1,H}(\pi) - \min_{\pi_H \in \Pi_{\mathrm{NM}}}\{F_{1,H}(\pi_H)\}}_{=\ \mathrm{OptGap}_H(\pi)} + 8L\gamma^H, \tag{10}$$

*where $\mathrm{OptGap}_H(\pi)$ is the optimality gap of policy $\pi$ under the single-trial truncated objective with horizon $H$.*

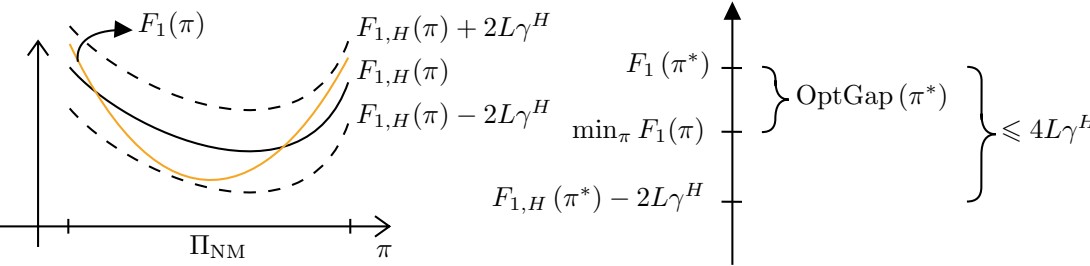

Figure 3: Illustration of objectives $F_1$ and $F_{1,H}$, as well as the relation between different quantities of interest for the proof.

*Proof.* Let $\pi_H^* = \arg\min_{\pi \in \Pi_{\text{NM}}} F_{1,H}(\pi)$, i.e., $\pi_H^*$ is optimal with respect to the truncated objective. It holds that,

$$
\begin{aligned}
\text{OptGap}(\pi) &= \mathbb{E}\left[f(\mathbf{d}^\pi)\right] - \min_{\pi' \in \Pi_{\text{NM}}} \mathbb{E}\left[f(\mathbf{d}^{\pi'})\right] \\
&= \left| \mathbb{E}\left[f(\mathbf{d}^\pi)\right] - \min_{\pi' \in \Pi_{\text{NM}}} \mathbb{E}\left[f(\mathbf{d}^{\pi'})\right] \right| \\
&\overset{(a)}{\leq} \left| \mathbb{E}\left[f(\mathbf{d}^\pi)\right] - \mathbb{E}\left[f(\mathbf{d}^{\pi_H^*})\right] \right| + \left| \mathbb{E}\left[f(\mathbf{d}^{\pi_H^*})\right] - \min_{\pi' \in \Pi_{\text{NM}}} \mathbb{E}\left[f(\mathbf{d}^{\pi'})\right] \right| \\
&\overset{(b)}{\leq} \left| \mathbb{E}\left[f(\mathbf{d}^\pi)\right] - \mathbb{E}\left[f(\mathbf{d}^{\pi_H^*})\right] \right| + 4L\gamma^H \\
&\overset{(c)}{\leq} \left| \mathbb{E}\left[f(\mathbf{d}^\pi)\right] - \mathbb{E}\left[f(\mathbf{d}^{\pi,H})\right] \right| + \left| \mathbb{E}\left[f(\mathbf{d}^{\pi,H})\right] - \mathbb{E}\left[f(\mathbf{d}^{\pi_H^*})\right] \right| + 4L\gamma^H \\
&\overset{(d)}{\leq} 2L\gamma^H + \left| \mathbb{E}\left[f(\mathbf{d}^{\pi,H})\right] - \mathbb{E}\left[f(\mathbf{d}^{\pi_H^*})\right] \right| + 4L\gamma^H \\
&\overset{(e)}{\leq} \left| \mathbb{E}\left[f(\mathbf{d}^{\pi,H})\right] - \mathbb{E}\left[f(\mathbf{d}^{\pi_H^*,H})\right] \right| + \left| \mathbb{E}\left[f(\mathbf{d}^{\pi_H^*,H})\right] - \mathbb{E}\left[f(\mathbf{d}^{\pi_H^*})\right] \right| + 6L\gamma^H \\
&\overset{(f)}{\leq} \left| \mathbb{E}\left[f(\mathbf{d}^{\pi,H})\right] - \mathbb{E}\left[f(\mathbf{d}^{\pi_H^*,H})\right] \right| + 8L\gamma^H \\
&= \mathbb{E}\left[f(\mathbf{d}^{\pi,H})\right] - \min_{\pi_H \in \Pi_{\text{NM}}} \left\{ \mathbb{E}\left[f(\mathbf{d}^{\pi_H,H})\right] \right\} + 8L\gamma^H \\
&= F_{1,H}(\pi) - \min_{\pi_H \in \Pi_{\text{NM}}} \left\{ F_{1,H}(\pi_H) \right\} + 8L\gamma^H
\end{aligned}
$$

where (a) follows from adding and subtracting $\mathbb{E}\left[f(\mathbf{d}^{\pi_H^*})\right]$ and applying the triangular inequality; (b) follows from Lemma 3; (c) follows from adding and subtracting $\mathbb{E}\left[f(\mathbf{d}^{\pi,H})\right]$ and applying the triangular inequality; (d) follows from Lemma 2; (e) follows from adding and subtracting $\mathbb{E}\left[f(\mathbf{d}^{\pi_H^*,H})\right]$ and applying the triangular inequality; and (f) follows from Lemma 2. $\qquad\square$

### B.3 THE OCCUPANCY MDP: VALUE AND ACTION-VALUE FUNCTIONS

For a given policy $\pi_O \in \Pi_S$, the interaction between the agent and the occupancy MDP gives rise to a random process $(\{s_0, \mathbf{o}_0\}, a_0, \{s_1, \mathbf{o}_1\}, a_1, \ldots, \{s_H, \mathbf{o}_H\})$ such that:

1. $\mathbb{P}\left[\{s_0, \mathbf{o}_0\} = \{s_0, \mathbf{o}_0\}\right] = p_{0,O}(\{s_0, \mathbf{o}_0\})$

2. $\mathbb{P}\left[\{s_{t+1}, \mathbf{o}_{t+1}\} = \{s', \mathbf{o}'\} | \{s_0, \mathbf{o}_0\}, a_0, \ldots, \{s_t, \mathbf{o}_t\}, a_t\right] = P_O^{a_t}(\{s_t, \mathbf{o}_t\}, \{s', \mathbf{o}'\})$

3. $\mathbb{P}\left[a_t = a | \{s_0, \mathbf{o}_0\}, a_0, \ldots, \{s_t, \mathbf{o}_t\}\right] = \pi_O(a | \{s_t, \mathbf{o}_t\})$

We let $(\Omega_O, \mathcal{F}_O, \mathbb{P}_{\pi_O}^O)$ be the probability space over the sequence of random variables $(\{s_0, \mathbf{o}_0\}, a_0, \{s_1, \mathbf{o}_1\}, a_1, \ldots, \{s_H, \mathbf{o}_H\})$ that satisfies conditions 1-3 above. We write specific trajectories as $\omega_O \in \Omega_O$, with $\omega_O = (\{s_0, \mathbf{o}_0\}, a_0, \{s_1, \mathbf{o}_1\}, a_1, \ldots, \{s_H, \mathbf{o}_H\})$. We highlight that the

probability of a given trajectory $\omega_O \in \Omega_O$ under stationary policy $\pi_O \in \Pi_S$ can be calculated as

$$\mathbb{P}^O_{\pi_O}(\omega_O) = p_{0,O}(\{s_0, \boldsymbol{o}_0\}) \cdot \pi_O(a_0|\{s_0, \boldsymbol{o}_0\}) \cdot P_O^{a_0}(\{s_0, \boldsymbol{o}_0\}, \{s_1, \boldsymbol{o}_1\}) \dots$$
$$\cdot \pi_O(a_{H-1}|\{s_{H-1}, \boldsymbol{o}_{H-1}\}) \cdot P_O^{a_{H-1}}(\{s_{H-1}, \boldsymbol{o}_{H-1}\}, \{s_H, \boldsymbol{o}_H\}).$$

To streamline our notation we introduce the mapping $\sigma : \mathcal{S} \times \mathcal{O} \times \mathcal{A} \to \mathcal{O}$ that describes the evolution of component $\boldsymbol{o}$ of the state, defined as

$$[\sigma(s, \boldsymbol{o}, a)]_{s', a'} = \begin{cases} o(s, a) + \gamma^t & \text{if } s' = s, a' = a, \\ o(s, a) & \text{otherwise,} \end{cases}$$

where $[o]_{s',a'}$ denotes the value of entry $s', a'$ for vector $\boldsymbol{o}$, i.e., $[o]_{s',a'} = o(s', a')$.

The *value function* under $\mathcal{M}_O$, for any $t \in \{0, \dots, H\}$, is defined as

$$V_t^{\pi_O}(\{s, \boldsymbol{o}\}) = \mathbb{E}_{\pi_O} \left[ \sum_{t'=t}^{H} c_O(\{s_{t'}, \boldsymbol{o}_{t'}\}) \middle| \{s_t, \boldsymbol{o}_t\} = \{s, \boldsymbol{o}\} \right] \tag{11}$$

$$= \mathbb{E}_{\pi_O} \left[ c_O(\{s_H, \boldsymbol{o}_H\}) \middle| \{s_t, \boldsymbol{o}_t\} = \{s, \boldsymbol{o}\} \right], \tag{12}$$

and the *optimal value function*, for any $t \in \{0, \dots, H\}$, as $V_t^*(\{s, \boldsymbol{o}\}) = \min_{\pi_O \in \Pi_S^p} V_t^{\pi_O}(\{s, \boldsymbol{o}\})$. The *action-value function* under $\mathcal{M}_O$, for any $t \in \{0, \dots, H-1\}$, is defined as

$$Q_t^{\pi_O}(\{s, \boldsymbol{o}\}, a) = \mathbb{E}_{\pi_O} \left[ \sum_{t'=t}^{H} c_O(\{s_{t'}, \boldsymbol{o}_{t'}\}) \middle| \{s_t, \boldsymbol{o}_t\} = \{s, \boldsymbol{o}\}, a_t = a \right]$$

$$= \mathbb{E}_{\pi_O} \left[ c_O(\{s_H, \boldsymbol{o}_H\}) \middle| \{s_t, \boldsymbol{o}_t\} = \{s, \boldsymbol{o}\}, a_t = a \right],$$

and the *optimal action-value function*, for any $t \in \{0, \dots, H-1\}$, as $Q_t^*(\{s, \boldsymbol{o}\}, a) = \min_{\pi_O \in \Pi_S^p} Q_t^{\pi_O}(\{s, \boldsymbol{o}\}, a)$. We emphasize again that subscript $t$ can be dropped from $V_t^{\pi_O}(\{s, \boldsymbol{o}\})$, $V_t^*(\{s, \boldsymbol{o}\})$, $Q_t^{\pi_O}(\{s, \boldsymbol{o}\}, a)$ and $Q_t^*(\{s, \boldsymbol{o}\}, a)$ as it can be inferred from $\boldsymbol{o}$. Finally, we note that value functions, optimal value functions and optimal action-value functions satisfy the following set of Bellman equations:

$$V_t^{\pi_O}(\{s, \boldsymbol{o}\}) = \sum_{a \in \mathcal{A}} \pi_O(a|\{s, \boldsymbol{o}\}) \left( \sum_{s' \in \mathcal{S}} P^a(s'|s) V_{t+1}^{\pi_O}(\{s', \sigma(s, \boldsymbol{o}, a)\}) \right), \quad \forall t \in \{0, \dots, H-1\}$$

$$V_t^*(\{s, \boldsymbol{o}\}) = \min_{a \in \mathcal{A}} \left\{ \sum_{s' \in \mathcal{S}} P^a(s'|s) V_{t+1}^*(\{s', \sigma(s, \boldsymbol{o}, a)\}) \right\}, \quad \forall t \in \{0, \dots, H-1\}$$

$$Q_t^*(\{s, \boldsymbol{o}\}, a) = \sum_{s' \in \mathcal{S}} P^a(s'|s) V_{t+1}^*(\{s', \sigma(s, \boldsymbol{o}, a)\}), \quad \forall t \in \{0, \dots, H-1\}$$

$$Q_t^*(\{s, \boldsymbol{o}\}, a) = \sum_{s' \in \mathcal{S}} P^a(s'|s) \min_{a' \in \mathcal{A}} \left\{ Q_{t+1}^*(\{s', \sigma(s, \boldsymbol{o}, a)\}, a') \right\}, \quad \forall t \in \{0, \dots, H-2\}.$$

### B.4 PROOF OF PROPOSITION 2

**Lemma 4** (Linear independence of exponential functions over $\mathbb{R}$). *For any* $x \in \mathbb{R}$, $L \in \mathbb{N}$, $c_0, \dots, c_{L-1} \in \mathbb{R}$, *and distinct* $\lambda_0, \dots, \lambda_{L-1} \in \mathbb{R}$, *if* $\sum_{t=0}^{L-1} c_t e^{\lambda_t x} = 0$ *then* $c_0 = c_1 = \dots = c_{L-1} = 0$, *i.e., the exponentials* $e^{\lambda_0 x}, \dots, e^{\lambda_{L-1} x}$ *are linearly independent over* $\mathbb{R}$.

*Proof.* Let $f(x) = \sum_{t=0}^{L-1} c_t e^{\lambda_t x}$. It holds that $\frac{d^k f}{dx^k} = \sum_{t=0}^{L-1} c_t \lambda_t^k e^{\lambda_t x}$. We can repeatedly differentiate $f$ to obtain the following set of equalities:

$$\sum_{t=0}^{L-1} c_t e^{\lambda_t x} = 0,$$

$$\sum_{t=0}^{L-1} c_t \lambda_t e^{\lambda_t x} = 0,$$

$$\sum_{t=0}^{L-1} c_t \lambda_t^2 e^{\lambda_t x} = 0,$$

$$\cdots$$

$$\sum_{t=0}^{L-1} c_t \lambda_t^{L-1} e^{\lambda_t x} = 0.$$

The set of equations above can be rearranged as follows

$$\begin{bmatrix} 1 & 1 & \cdots & 1 \\ \lambda_0 & \lambda_1 & \cdots & \lambda_{L-1} \\ \lambda_0^2 & \lambda_1^2 & \cdots & \lambda_{L-1}^2 \\ \vdots & \vdots & \ddots & \vdots \\ \lambda_0^{L-1} & \lambda_1^{L-1} & \cdots & \lambda_{L-1}^{L-1} \end{bmatrix} \begin{bmatrix} c_0 e^{\lambda_0 x} \\ c_1 e^{\lambda_1 x} \\ c_2 e^{\lambda_2 x} \\ \vdots \\ c_{L-1} e^{\lambda_{L-1} x} \end{bmatrix} = \begin{bmatrix} 0 \\ 0 \\ 0 \\ \vdots \\ 0 \end{bmatrix}. \tag{13}$$

The square matrix above is known as the Vandermonde matrix and, since all $\lambda_0, \ldots, \lambda_{L-1}$ are distinct, the matrix has a non-zero determinant (hence it is invertible). Therefore, multiplying the equality above by the inverse of the Vandermonde matrix on the left we obtain

$$\begin{bmatrix} c_0 e^{\lambda_0 x} \\ c_1 e^{\lambda_1 x} \\ c_2 e^{\lambda_2 x} \\ \vdots \\ c_{L-1} e^{\lambda_{L-1} x} \end{bmatrix} = \begin{bmatrix} 0 \\ 0 \\ 0 \\ \vdots \\ 0 \end{bmatrix}. \tag{14}$$

Since functions $e^{\lambda_0 x}, \ldots, e^{\lambda_{L-1} x}$ are always positive, we have that $c_0 = c_1 = \ldots = c_{L-1} = 0$. $\quad\square$

**Proposition 2** (One-to-one mapping between histories in $\mathcal{M}_f$ and states in $\mathcal{M}_O$)**.** *There exists a one-to-one mapping between histories $h_l = (s_0, a_0, s_1, a_1, \ldots, s_l) \in \mathcal{S} \times (\mathcal{S} \times \mathcal{A})^l$ in $\mathcal{M}_f$, with $0 \le l \le H - 1$, and states $\{s, \boldsymbol{o}\} \in \mathcal{S} \times \mathcal{O}$ in $\mathcal{M}_O$.*

*Proof.* For a given history $h_l = (s_0, a_0, s_1, a_1, \ldots, s_l) \in \mathcal{S} \times (\mathcal{S} \times \mathcal{A})^l$ in $\mathcal{M}_f$, with $0 \le l \le H-1$, consider the mapping defined below that associates $h_l$ to a given state $\{s, \boldsymbol{o}\} \in \mathcal{S} \times \mathcal{O}$ for $\mathcal{M}_O$ by letting

$$s = s_l \quad \text{and} \quad o(s, a) = \sum_{t=0}^{l-1} \gamma^t \mathbf{1}(s_t = s, a_t = a), \ \forall s \in \mathcal{S}, a \in \mathcal{A}. \tag{15}$$

We aim to show that the mapping above is a bijection between the set of possible histories in $\mathcal{M}_f$ and the discrete state space $\mathcal{O}$ in $\mathcal{M}_O$. Clearly, from the mapping above defined, each history $h_l$ in $\mathcal{M}_f$ is associated with a unique state in $\mathcal{M}_O$. Thus, what remains is to show that any two states $\{s_1, \boldsymbol{o}_1\}$ and $\{s_2, \boldsymbol{o}_2\}$ for $\mathcal{M}_O$ are equal under mapping equation 15 if and only if their associated histories $h^1$ and $h^2$ are equal. We now make two observations. First, for a given state $\{s, \boldsymbol{o}\}$ in $\mathcal{M}_O$, component $s$ is directly related, through mapping equation 15, to the last state in the history $h_l$. Second, each history $h_l = (s_0, a_0, s_1, a_1, \ldots, s_l)$ will yield through mapping equation 15 a running occupancy $\boldsymbol{o}$ satisfying $\sum_{s,a} o(s, a) = \frac{1-\gamma^l}{1-\gamma}$; thus, histories $h_l$ with different lengths will yield different $\boldsymbol{o}$-vectors. Hence, we only need to show that two running occupancies $\boldsymbol{o}_1$ and $\boldsymbol{o}_2$, associated with histories $h^1$ and $h^2$ (both of length $l$), respectively, are the same if and only if their histories up to timestep $l - 1$ are the same:

- If two histories $h^1$ and $h^2$ of length $l$ are the same, then it should be clear that their respective running occupancies, as defined through equation 15, are also the same.

- If two running occupancies $\boldsymbol{o}_1$ and $\boldsymbol{o}_2$ are the same, then their associated histories are also the same. To prove this implication, we focus our attention to a given entry $(s, a)$ of the vectors $\boldsymbol{o}_1$ and $\boldsymbol{o}_2$. Running occupancy $\boldsymbol{o}_1$ is associated with an arbitrary history $h^1 = (s_0^1, a_0^1, s_1^1, a_1^1, \ldots, s_l^1)$; running occupancy $\boldsymbol{o}_2$ is associated with an arbitrary history $h^2 = (s_0^2, a_0^2, s_1^2, a_1^2, \ldots, s_l^2)$. If $\boldsymbol{o}_1 = \boldsymbol{o}_2$ then, for any $s \in \mathcal{S}, a \in \mathcal{A}$,

$$o_1(s, a) - o_2(s, a) = 0$$

$$\Leftrightarrow \quad \sum_{t=0}^{l-1} \gamma^t \mathbf{1}(s_t^1 = s, a_t^1 = a) - \sum_{t=0}^{l-1} \gamma^t \mathbf{1}(s_t^2 = s, a_t^2 = a) = 0$$

$$\Leftrightarrow \quad \sum_{t=0}^{l-1} \gamma^t \left( \mathbf{1}(s_t^1 = s, a_t^1 = a) - \mathbf{1}(s_t^2 = s, a_t^2 = a) \right) = 0$$

$$\overset{(a)}{\Leftrightarrow} \quad \sum_{t=0}^{l-1} \gamma^t c_t = 0,$$

where in (a) we let $c_t \in \{-1, 0, 1\}$. Now, the only solution to the last equation above is $c_0 = c_1 = \ldots = c_{l-1} = 0$, which implies that $\mathbf{1}(s_t^1 = s, a_t^1 = a) = \mathbf{1}(s_t^2 = s, a_t^2 = a)$ for all $o \leq t \leq l-1$ and hence, the histories are the same. The fact that $c_0 = c_1 = \ldots = c_{l-1} = 0$ is the only solution to the equation above follows from Lemma 4 by letting $L = l$, $x = 1$, and $\lambda_t = \ln(\gamma)t$ (which implies that all $\lambda_t$ are distinct for $\gamma \in (0, 1)$).

Thus, we conclude that there exists a one-to-one mapping, as defined in equation 15, between every possible history in $\mathcal{M}_f$ up to timestep $H - 1$ and states in $\mathcal{M}_O$. $\qquad \square$

### B.5 PROOF OF THEOREM 2

**Theorem 2** (Solving $\mathcal{M}_f$ is "equivalent" to solving $\mathcal{M}_O$). *The problem of finding a policy $\pi \in \Pi_{NM}$ satisfying $\mathrm{OptGap}_H(\pi) \leq \epsilon$, for any $\epsilon \in \mathbb{R}_0^+$, can be reduced to the problem of finding a policy $\pi_O \in \Pi_S$ satisfying $J_O(\pi_O) - J_O^* \leq \epsilon$. In particular, if $\pi_O^* = \arg\min_{\pi_O \in \Pi_S} J_O(\pi_O)$, then the corresponding non-Markovian policy $\pi$ in $\mathcal{M}_f$ satisfies $\mathrm{OptGap}_H(\pi) = 0$. Finally, it holds that $\mathrm{OptGap}_H(\pi) = J_O(\pi_O) - J_O^*$, where $\pi_O$ is the stationary policy for $\mathcal{M}_O$ associated with the non-Markovian policy $\pi$ for $\mathcal{M}_f$.*

*Proof.* We start by showing that, for any horizon $H \in \mathbb{N}$ and policy $\pi \in \Pi_{NM}$, it holds that

$$F_{1,H}(\pi) = J_O(\pi_O),$$

for $F_{1,H}(\pi)$ as defined in equation 6 and $J_O(\pi_O)$ as defined in equation 9, where $\pi_O$ is the stationary policy for $\mathcal{M}_O$ associated with the non-Markovian policy $\pi$ for $\mathcal{M}_f$.

For any $H \in \mathbb{N}$, finite-horizon random trajectories $(\mathrm{s}_0, \mathrm{a}_0, \mathrm{s}_1, \mathrm{a}_1, \ldots, \mathrm{s}_{H-1}, \mathrm{a}_{H-1})$ in $\mathcal{M}_f$ are associated with the probability space $(\Omega, \mathcal{F}, \mathbb{P}_\pi)$. We write specific trajectories as $\omega \in \Omega$, with $\omega = (s_0, a_0, s_1, a_1, \ldots, s_{H-1}, a_{H-1})$. We highlight that the probability of a given trajectory $\omega \in \Omega$ under policy $\pi \in \Pi_{NM}$ can be calculated as $\mathbb{P}_\pi[\omega] = p_0(s_0) \cdot \pi(a_0|h_0) \cdot P^{a_0}(s_0, s_1) \cdot \pi(a_1|h_1) \cdot P^{a_1}(s_1, s_2) \ldots P^{a_{H-2}}(s_{H-2}, s_{H-1}) \cdot \pi(a_{H-1}|h_{H-1})$. On the other hand, random trajectories $(\{\mathrm{s}_0, \mathbf{o}_0\}, \mathrm{a}_0, \{\mathrm{s}_1, \mathbf{o}_1\}, \mathrm{a}_1, \ldots, \{\mathrm{s}_H, \mathbf{o}_H\})$ in $\mathcal{M}_O$ are associated with probability space $(\Omega_O, \mathcal{F}_O, \mathbb{P}_{\pi_O}^O)$. We write specific trajectories as $\omega_O \in \Omega_O$, with $\omega_O = (\{s_0, \mathbf{o}_0\}, a_0, \{s_1, \mathbf{o}_1\}, a_1, \ldots, \{s_H, \mathbf{o}_H\})$.

We start by noting that, for any trajectory $\omega_O = (\{s_0, \mathbf{o}_0\}, a_0, \{s_1, \mathbf{o}_1\}, a_1, \ldots, \{s_H, \mathbf{o}_H\}) \in \Omega_O$,

$$\mathbb{P}_{\pi_O}^O[\omega_O] = p_{0,O}(\{s_0, \mathbf{o}_0\}) \cdot \pi_O(a_0|\{s_0, \mathbf{o}_0\}) \cdot P_O^{a_0}(\{s_0, \mathbf{o}_0\}, \{s_1, \mathbf{o}_1\}) \cdot \ldots$$
$$\cdot \pi_O(a_{H-1}|\{s_{H-1}, \mathbf{o}_{H-1}\}) \cdot P_O^{a_{H-1}}(\{s_{H-1}, \mathbf{o}_{H-1}\}, \{s_H, \mathbf{o}_H\}).$$

$$\overset{(a)}{=} p_0(s_0) \cdot \mathbf{1}(\mathbf{o}_0 = [0, \ldots, 0]) \cdot \pi_O(a_0|\{s_0, \mathbf{o}_0\}) \cdot P^{a_0}(s_0, s_1) \cdot \mathbf{1}(\mathbf{o}_1 = \sigma(s_0, \mathbf{o}_0, a_0)) \cdot \ldots$$
$$\cdot \pi_O(a_{H-1}|\{s_{H-1}, \mathbf{o}_{H-1}\}) \cdot P^{a_{H-1}}(s_{H-1}, s_H) \cdot \mathbf{1}(\mathbf{o}_H = \sigma(s_{H-1}, \mathbf{o}_{H-1}, a_{H-1}))$$

$$\overset{(b)}{=} p_0(s_0) \cdot \mathbf{1}(\mathbf{o}_0 = [0, \ldots, 0]) \cdot \pi(a_0|h_0) \cdot P^{a_0}(s_0, s_1) \cdot \mathbf{1}(\mathbf{o}_1 = \sigma(s_0, \mathbf{o}_0, a_0)) \cdot \ldots$$
$$\cdot \pi(a_{H-1}|h_{H-1}) \cdot P^{a_{H-1}}(s_{H-1}, s_H) \cdot \mathbf{1}(\mathbf{o}_H = \sigma(s_{H-1}, \mathbf{o}_{H-1}, a_{H-1}))$$

$$\overset{(c)}{=} \mathbb{P}_\pi[\omega] \cdot P^{a_{H-1}}(s_{H-1}, s_H) \cdot \mathbf{1}(\mathbf{o}_0 = [0, \ldots, 0]) \cdot \mathbf{1}(\mathbf{o}_1 = \sigma(s_0, \mathbf{o}_0, a_0)) \cdot \ldots$$
$$\cdot \mathbf{1}(\mathbf{o}_H = \sigma(s_{H-1}, \mathbf{o}_{H-1}, a_{H-1})),$$

where in (a) we note that component $\mathbf{o}$ of the state is initialized as a zero vector and then deterministically evolves according to $\sigma$; any sequence of $\mathbf{o}$-vectors that does not evolve according to $\sigma$ has

zero probability under probability measure $\mathbb{P}^{\mathrm{O}}_{\pi_\mathrm{O}}$. In (b) we used the fact that any stationary policy $\pi_\mathrm{O} \in \Pi_\mathrm{S}$ for $\mathcal{M}_\mathrm{O}$ can be mapped to a particular non-Markovian policy $\pi \in \Pi_\mathrm{NM}$ in $\mathcal{M}_f$. In (c) we recall that, for $\omega = (s_0, a_0, s_1, a_1, \dots, s_{H-1}, a_{H-1})$, $\mathbb{P}_\pi[\omega] = p_0(s_0) \cdot \pi(a_0|h_0) \cdot P^{a_0}(s_0, s_1) \cdot \pi(a_1|h_1) \cdot P^{a_1}(s_1, s_2) \dots P^{a_{H-2}}(s_{H-2}, s_{H-1}) \cdot \pi(a_{H-1}|h_{H-1})$.

Now, for any stationary policy $\pi_\mathrm{O} \in \Pi_\mathrm{S}$, it holds that

$$
\begin{aligned}
J_\mathrm{O}(\pi_\mathrm{O}) &= \mathbb{E}\left[c_\mathrm{O}(\{s_H, \boldsymbol{o}_H\})\right] \\
&= \sum_{\omega_\mathrm{O} \in \Omega_\mathrm{O}} \mathbb{P}^{\mathrm{O}}_{\pi_\mathrm{O}}[\omega_\mathrm{O}] c_\mathrm{O}(\{s_H, \boldsymbol{o}_H\}) \\
&= \sum_{\omega_\mathrm{O} \in \Omega_\mathrm{O}} \mathbb{P}_\pi[\omega] \cdot P^{a_{H-1}}(s_{H-1}, s_H) \cdot \mathbf{1}(\boldsymbol{o}_0 = [0, \dots, 0]) \cdot \mathbf{1}(\boldsymbol{o}_1 = \sigma(s_0, \boldsymbol{o}_0, a_0)) \cdot \dots \\
&\qquad \cdot \mathbf{1}(\boldsymbol{o}_H = \sigma(s_{H-1}, \boldsymbol{o}_{H-1}, a_{H-1})) f\left(\frac{1-\gamma}{1-\gamma^H} \boldsymbol{o}_H\right) \\
&\overset{(a)}{=} \sum_{\omega_\mathrm{O} \in \Omega_\mathrm{O}} \mathbb{P}_\pi[\omega] \cdot P^{a_{H-1}}(s_{H-1}, s_H) \cdot \mathbf{1}(\boldsymbol{o}_0 = [0, \dots, 0]) \cdot \mathbf{1}(\boldsymbol{o}_1 = \sigma(s_0, \boldsymbol{o}_0, a_0)) \cdot \dots \\
&\qquad \cdot \mathbf{1}(\boldsymbol{o}_H = \sigma(s_{H-1}, \boldsymbol{o}_{H-1}, a_{H-1})) f\left(\mathbf{d}^{\pi,H}_\omega\right) \\
&\overset{(b)}{=} \sum_{\omega \in \Omega} \mathbb{P}_\pi[\omega] f\left(\mathbf{d}^{\pi,H}_\omega\right) \sum_{\boldsymbol{o}_0, \boldsymbol{o}_1, \dots, \boldsymbol{o}_H \in \mathcal{O}} \sum_{s_H \in \mathcal{S}} P^{a_{H-1}}(s_{H-1}, s_H) \cdot \\
&\qquad \mathbf{1}(\boldsymbol{o}_0 = [0, \dots, 0]) \cdot \mathbf{1}(\boldsymbol{o}_1 = \sigma(s_0, \boldsymbol{o}_0, a_0)) \cdot \dots \cdot \mathbf{1}(\boldsymbol{o}_H = \sigma(s_{H-1}, \boldsymbol{o}_{H-1}, a_{H-1})) \\
&\overset{(c)}{=} \sum_{\omega \in \Omega} \mathbb{P}_\pi[\omega] f\left(\mathbf{d}^{\pi,H}_\omega\right) \\
&= \mathbb{E}\left[f(\mathbf{d}^{\pi,H})\right] \\
&= F_{1,H}(\pi),
\end{aligned}
$$

where in (a) we noted that, for any $\omega_\mathrm{O} \in \Omega_\mathrm{O}$, $f\left(\frac{1-\gamma}{1-\gamma^H} \boldsymbol{o}_H\right) = f\left(\mathbf{d}^{\pi,H}_\omega\right)$. In (b), we split the sum over $\omega_\mathrm{O} \in \Omega_\mathrm{O}$ as a sum over $\omega \in \Omega$, a sum over each possible vector $\boldsymbol{o} \in \mathcal{O}$ across all timesteps, and a sum over the final state $s_H \in \mathcal{S}$ (not included in $\omega$). We also rearranged the sums by noting that some terms do not depend on some of the sums. In (c) we note that the inner sums over the $\boldsymbol{o}$-vectors and $s_H$ equal one.

Hence, we have proven that, for any $H \in \mathbb{N}$ and $\pi \in \Pi_\mathrm{NM}$, $F_{1,H}(\pi) = J_\mathrm{O}(\pi_\mathrm{O})$ holds, where, in light of Prop. 2, $\pi_\mathrm{O}$ is the stationary policy for $\mathcal{M}_\mathrm{O}$ associated with the non-Markovian policy $\pi$ for $\mathcal{M}_f$. Given this result, and due to the one-to-one mapping between non-Markovian policies for $\mathcal{M}_f$ and stationary policies for $\mathcal{M}_\mathrm{O}$, it holds for any $\pi \in \Pi_\mathrm{NM}$ that

$$
\mathrm{OptGap}_H(\pi) = F_{1,H}(\pi) - \min_{\pi_H \in \Pi_\mathrm{NM}} \{F_{1,H}(\pi_H)\} = J_\mathrm{O}(\pi_\mathrm{O}) - \min_{\pi'_\mathrm{O} \in \Pi_\mathrm{S}} J_\mathrm{O}(\pi'_\mathrm{O}),
$$

and the conclusion follows. □

### B.6 Proof of Theorem 3

**Theorem 3** (NP-Hardness of policy optimization in the single-trial regime). *Given a GUMDP with objective $F_{1,H}$ and a threshold value $\lambda \in \mathbb{R}$, it is NP-Hard to determine whether there exists a policy $\pi \in \Pi^{\mathrm{D}}_\mathrm{NM}$ satisfying $F_{1,H}(\pi) \leq \lambda$.*

*Proof.* We reduce the subset sum problem to the policy existence problem in GUMDPs with objective $F_{1,H}$. The subset sum problem asks, given a set $\mathcal{N} = \{n_0, n_1, \dots, n_{N-1}\}$ of $N$ non-negative integer numbers and a target sum $k \in \mathbb{N}$, whether there exists a subset of the numbers such that the sum of the elements in the set is $k$. The policy existence problem is: given a GUMDP with objective $F_{1,H}$ and a threshold value $\lambda \in \mathbb{R}$, does there exist a policy $\pi \in \Pi^{\mathrm{D}}_\mathrm{NM}$ such that $F_{1,H}(\pi) \leq \lambda$.

We map every instance of the subset sum problem as a GUMDP as follows: (i) the state space is $\mathcal{S} = \{s_0, s_1, \dots, s_N\}$; (ii) the action space is $\mathcal{A} = \{a_\mathrm{include}, a_\mathrm{not\text{-}include}\}$; (iii) $P^a(s_{i+1}|s_i) = 1$ and

zero otherwise for any $a \in \mathcal{A}$ and $i \in \{0, \ldots, N-1\}$, and $P^a(s_N|s_N) = 1$ for any $a \in \mathcal{A}$; (iv) $p_0(s_0) = 1$ and zero otherwise. We provide an illustration of the GUMDP in Fig. 4. We describe a discounted occupancy for GUMDP above defined with the vector

$$\boldsymbol{d} = [d(s_0, a_{\text{include}}), d(s_0, a_{\text{not-include}}), d(s_1, a_{\text{include}}), d(s_1, a_{\text{not-include}}), \ldots,$$
$$d(s_{N-1}, a_{\text{include}}), d(s_{N-1}, a_{\text{not-include}}), d(s_N, a_{\text{include}}), d(s_N, a_{\text{not-include}})].$$

Then, we set $H \geq N$ and let $f(\boldsymbol{d}) = (\boldsymbol{n}^\top \boldsymbol{d} - k)^2$, where

$$\boldsymbol{n} = \left[ \frac{1-\gamma^H}{1-\gamma} n_0, 0, \frac{1-\gamma^H}{(1-\gamma)\gamma} n_1, 0, \ldots, \frac{1-\gamma^H}{(1-\gamma)\gamma^{N-1}} n_{N-1}, 0, 0, 0 \right].$$

It holds that

$$\min_{\pi \in \Pi_{\text{NM}}^{\text{D}}} F_{1,H}(\pi) = \min_{\pi \in \Pi_{\text{NM}}^{\text{D}}} \mathbb{E}\left[ f(\mathbf{d}^{\pi,H}) \right] = \sum_{\omega \in \Omega} \mathbb{P}[\omega] f(\mathbf{d}_\omega^{\pi,H}).$$

For a given policy $\pi \in \Pi_{\text{NM}}^{\text{D}}$, only one trajectory $\omega \in \Omega$ has non-zero probability. The vector $\mathbf{d}_\omega^{\pi,H}$ associated with such a trajectory can be described as follows, for any $s_i \in \{0, \ldots, N-1\}$: (i) if at state $s_i$, $\pi$ selects $a_{\text{include}}$ then entry $\mathrm{d}_\omega^{\pi,H}(s_i, a_{\text{include}}) = \frac{1-\gamma}{1-\gamma^H}\gamma^i$ and $\mathrm{d}_\omega^{\pi,H}(s_i, a_{\text{not-include}}) = 0$; (ii) if at state $s_i$, $\pi$ selects $a_{\text{not-include}}$ then entry $\mathrm{d}_\omega^{\pi,H}(s_i, a_{\text{include}}) = 0$ and entry $\mathrm{d}_\omega^{\pi,H}(s_i, a_{\text{not-include}}) = \frac{1-\gamma}{1-\gamma^H}\gamma^i$. The action selected at $s_N$ is irrelevant since it does not affect the objective value. The intuition behind the GUMDP above defined is that, at each state $s_i$ for $i \in \{0, \ldots, N-1\}$, the policy needs to decide on whether to select action $a_{\text{include}}$ and, therefore, include term $n_i$ in the sum, or to select action $a_{\text{not-include}}$ and, therefore, not include term $n_i$ in the sum. We build the vector $\boldsymbol{n}$ to reflect such behavior, where each entry in $\boldsymbol{n}$ associated with the state $s_i$ and action $a_{\text{include}}$ has a normalizing constant of $\frac{1-\gamma^H}{(1-\gamma)\gamma^i}$ to account for the fact that the occupancy is discounted, as introduced in equation 4. Thus, it can be seen that every policy $\pi \in \Pi_{\text{NM}}^{\text{D}}$ will induce a particular trajectory $\omega \in \Omega$ with probability one and the sum of the numbers selected by the policy is given by $\boldsymbol{n}^\top \mathbf{d}_\omega^{\pi,H}$. Finally, the objective is such that $f(\boldsymbol{d}) = 0$ if and only if the sum of the selected numbers equals $k$. The policy existence problem then asks whether there exists a policy $\pi \in \Pi_{\text{NM}}^{\text{D}}$ such that $F_{1,H}(\pi) \leq \lambda$. By setting $\lambda = 0$ we are asking whether there exists a policy such that $F_{1,H}(\pi) \leq 0$. Since $f(\boldsymbol{d}) = 0$ if and only if the sum of the selected numbers equals $k$, we completed our reduction from the subset problem to the policy existence problem in GUMDPs with objective $F_{1,H}$. □

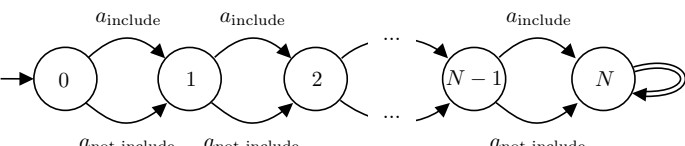

Figure 4: GUMDP instance used in the NP-Hardness proof.

## C SUPPLEMENTARY MATERIALS FOR SEC. 5

### C.1 TASKS AND ENVIRONMENTS

We consider three tasks: (i) maximum state entropy exploration (Hazan et al., 2019), where the objective is to visit all state-action pairs as uniformly as possible; imitation learning (Abbeel & Ng, 2004), where the objective is to imitate a given behaviour policy; and (iii) adversarial MDPs (Rosenberg & Mansour, 2019), where an adversary player selects the cost function that yields the highest cost. We refer to Sec. 2.3 for the exact definition of the objective function for each of these tasks. We normalize all objective functions to lie in the $[0, 1]$ interval. We consider two sets of environments. The first set corresponds to the illustrative GUMDPs depicted in Fig. 1, each associated with one of the tasks. The second set of environments come from the OpenAI Gym library (Brockman et al., 2016). We consider the FrozenLake (FL), the Taxi, and the Mountaincar (MC) environments. For the MC environment, we partitioned the original state space using equally spaced bins (we consider 10 bins per dimension). For the FL, Taxi and MC environments, the task of imitation learning consists in imitating and approximately optimal policy.

## C.2 EXPERIMENTAL METHODOLOGY, BASELINES, AND HYPERPARAMETERS

We perform 10 runs per experimental setting and report the 90% bootstrapped confidence interval. We let $\gamma = 0.9$. We consider two baselines. The first baseline is the random policy, $\pi_{\text{Random}}$. The second baseline is the optimal policy for the infinite trials formulation equation 2, $\pi^*_{\text{Solver}}$, that we calculate by solving a constrained optimization problem with objective $f$ using the Gurobi optimizer (Gurobi Optimization, LLC, 2025). More precisely, to compute $\pi^*_{\text{Solver}}$ we first solve the following optimization problem:

$$\boldsymbol{d}^* = \arg\min_{\boldsymbol{d} \in \mathcal{D}} f(\boldsymbol{d}),$$

$$\mathcal{D} = \{\boldsymbol{d} \in \mathbb{R}^{|\mathcal{S}||\mathcal{A}|} : d(s,a) \geq 0 \ \forall s,a, \sum_a d(s,a) = (1-\gamma)p_0(s) + \gamma \sum_{s',a} P^a(s|s')d(s',a) \ \forall s\}$$

Then, we let $\pi^*_{\text{Solver}}(a|s) = d^*(s,a)/\sum_{a'} d^*(s,a')$. For the illustrative GUMDPs we directly use the respective initial states distribution, $p_0$, and the transition probablity matrix, $P^a$. For the case of the OpenAI Gym environments we run a samling procedure to first estimate $p_0$ and $P^a$, and then we feed the estimated quantities to the optimization solver.

We denote the policy induced by our MCTS algorithm as $\pi_{\text{MCTS}}$. We use 4000 as the default number of iterations of the MCTS algorithm, but we also provide results for $10, 20, 50, 100, 500, 1000, 2000, 3000, 4000$ iteration steps. We submit our code in the zip file with our submission.

Our experiments required modest computational resources, with each experimental setting running in under an hour on a CPU cluster.

## C.3 COMPLETE EXPERIMENTAL RESULTS

### C.3.1 MAXIMUM STATE ENTROPY EXPLORATION, $\mathcal{M}_{f,1}$

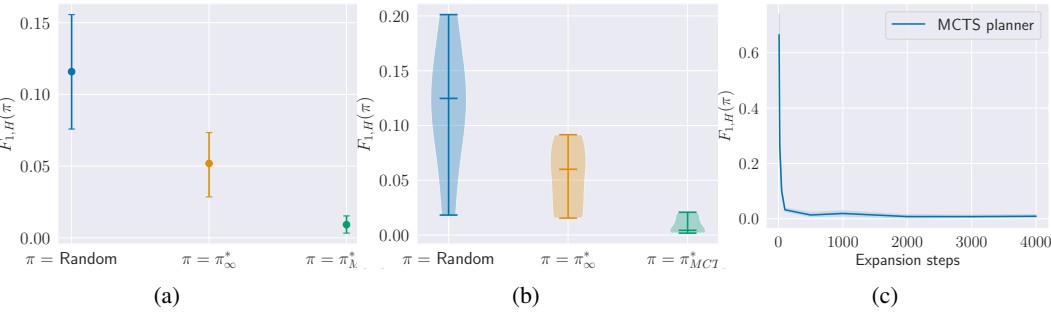

Figure 5: Maximum state entropy exploration, $\mathcal{M}_{f,1}$: (a) - Mean single-trial objective $F_{1,H}(\pi)$ obtained by different policies. Error bars correspond to the 90% mean confidence interval. (b) - Distribution of the single-trial objective $F_{1,H}(\pi)$ obtained by different policies. (c) - Mean single-trial objective $F_{1,H}(\pi)$ obtained by the MCTS-based algorithm as a function of the number of expansion steps. Shaded areas correspond to the 90% mean confidence interval. Across all plots, lower is better.

### C.3.2 Maximum state entropy exploration, FrozenLake

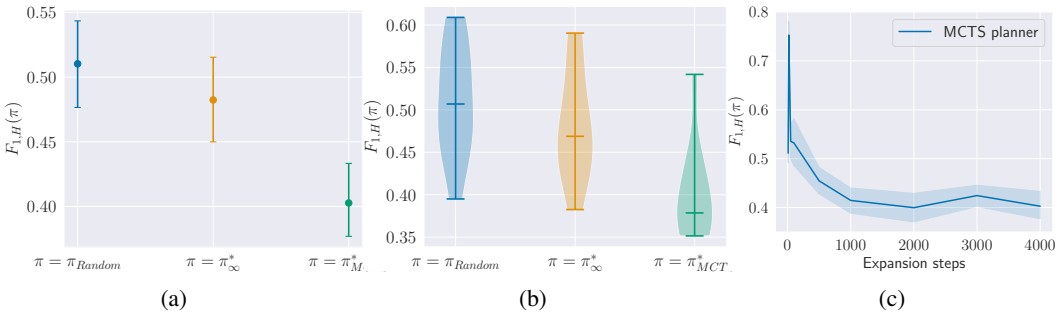

(a)  (b)  (c)

Figure 6: Maximum state entropy exploration, FrozenLake: (a) - Mean single-trial objective $F_{1,H}(\pi)$ obtained by different policies. Error bars correspond to the $90\%$ mean confidence interval. (b) - Distribution of the single-trial objective $F_{1,H}(\pi)$ obtained by different policies. (c) - Mean single-trial objective $F_{1,H}(\pi)$ obtained by the MCTS-based algorithm as a function of the number of expansion steps. Shaded areas correspond to the $90\%$ mean confidence interval. Across all plots, lower is better.

### C.3.3 Maximum state entropy exploration, Taxi

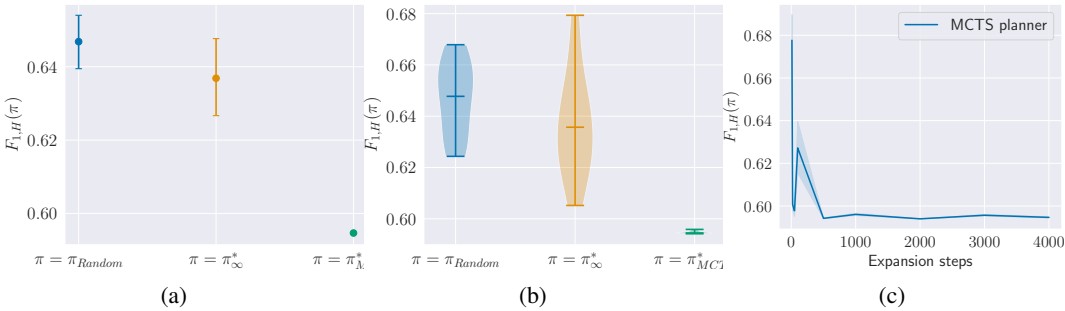

(a)  (b)  (c)

Figure 7: Maximum state entropy exploration, Taxi: (a) - Mean single-trial objective $F_{1,H}(\pi)$ obtained by different policies. Error bars correspond to the $90\%$ mean confidence interval. (b) - Distribution of the single-trial objective $F_{1,H}(\pi)$ obtained by different policies. (c) - Mean single-trial objective $F_{1,H}(\pi)$ obtained by the MCTS-based algorithm as a function of the number of expansion steps. Shaded areas correspond to the $90\%$ mean confidence interval. Across all plots, lower is better.

### C.3.4 MAXIMUM STATE ENTROPY EXPLORATION, MOUNTAINCAR

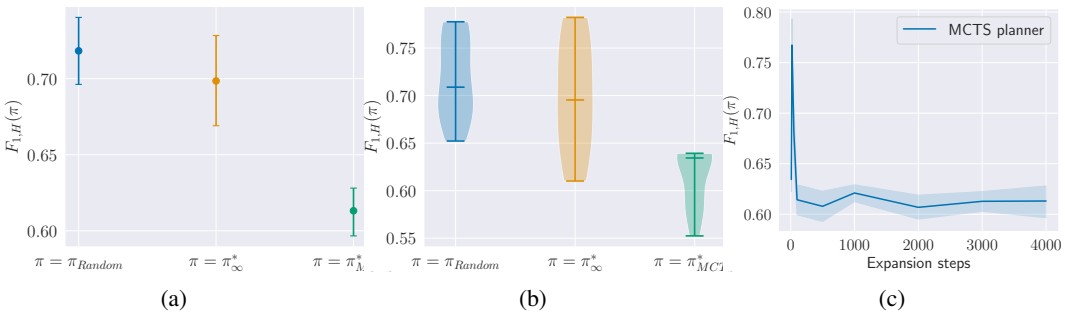

Figure 8: Maximum state entropy exploration, Mountaincar: (a) - Mean single-trial objective $F_{1,H}(\pi)$ obtained by different policies. Error bars correspond to the $90\%$ mean confidence interval. (b) - Distribution of the single-trial objective $F_{1,H}(\pi)$ obtained by different policies. (c) - Mean single-trial objective $F_{1,H}(\pi)$ obtained by the MCTS-based algorithm as a function of the number of expansion steps. Shaded areas correspond to the $90\%$ mean confidence interval. Across all plots, lower is better.

### C.3.5 IMITATION LEARNING, $\mathcal{M}_{f,2}$

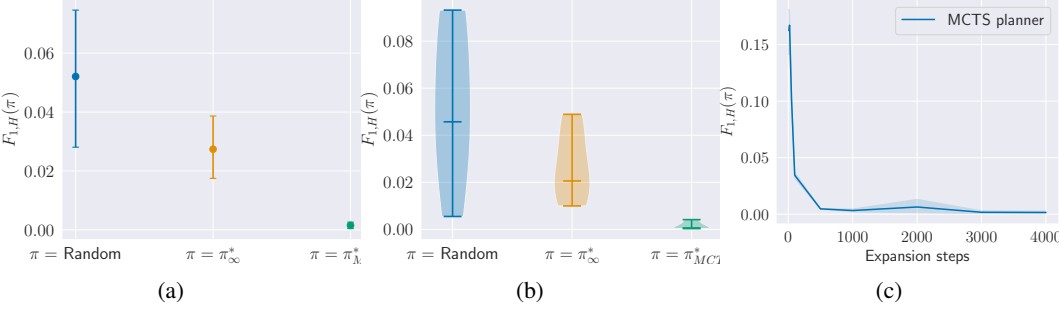

Figure 9: Imitation learning, $\mathcal{M}_{f,2}$: (a) - Mean single-trial objective $F_{1,H}(\pi)$ obtained by different policies. Error bars correspond to the $90\%$ mean confidence interval. (b) - Distribution of the single-trial objective $F_{1,H}(\pi)$ obtained by different policies. (c) - Mean single-trial objective $F_{1,H}(\pi)$ obtained by the MCTS-based algorithm as a function of the number of expansion steps. Shaded areas correspond to the $90\%$ mean confidence interval. Across all plots, lower is better.

### C.3.6 IMITATION LEARNING, FROZENLAKE

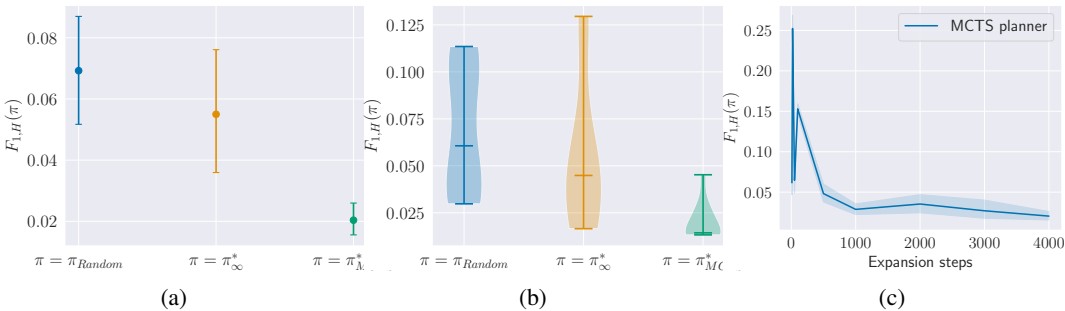

(a) (b) (c)

Figure 10: Imitation learning, FrozenLake: (a) - Mean single-trial objective $F_{1,H}(\pi)$ obtained by different policies. Error bars correspond to the $90\%$ mean confidence interval. (b) - Distribution of the single-trial objective $F_{1,H}(\pi)$ obtained by different policies. (c) - Mean single-trial objective $F_{1,H}(\pi)$ obtained by the MCTS-based algorithm as a function of the number of expansion steps. Shaded areas correspond to the $90\%$ mean confidence interval. Across all plots, lower is better.

### C.3.7 IMITATION LEARNING, TAXI

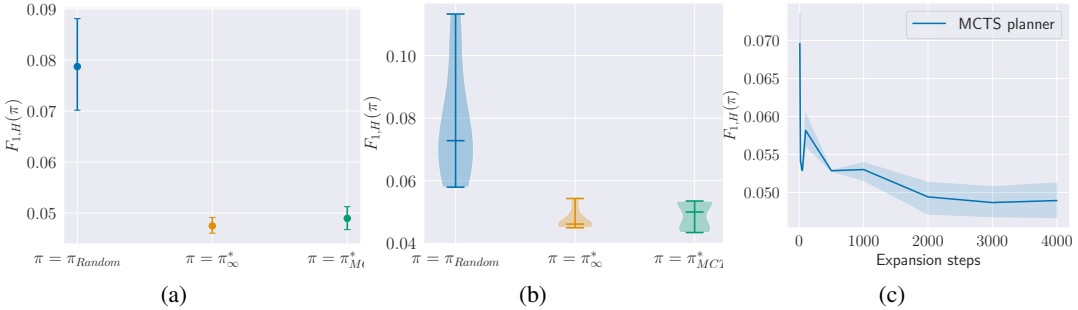

(a) (b) (c)

Figure 11: Imitation learning, Taxi: (a) - Mean single-trial objective $F_{1,H}(\pi)$ obtained by different policies. Error bars correspond to the $90\%$ mean confidence interval. (b) - Distribution of the single-trial objective $F_{1,H}(\pi)$ obtained by different policies. (c) - Mean single-trial objective $F_{1,H}(\pi)$ obtained by the MCTS-based algorithm as a function of the number of expansion steps. Shaded areas correspond to the $90\%$ mean confidence interval. Across all plots, lower is better.

### C.3.8   Imitation learning, MountainCar

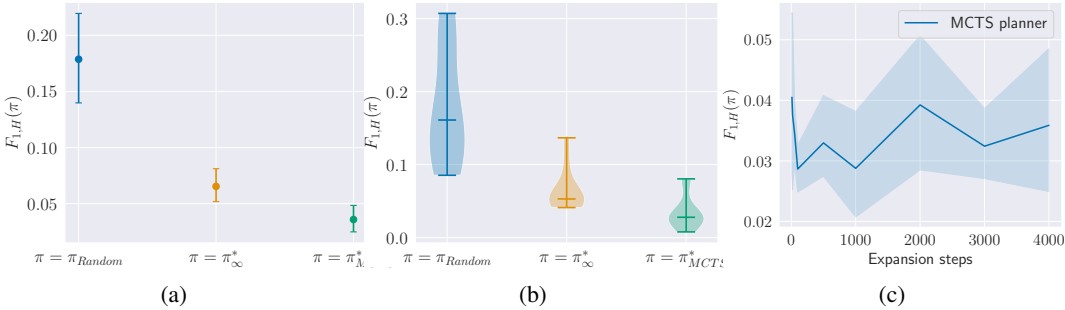

Figure 12: Imitation learning, MountainCar: (a) - Mean single-trial objective $F_{1,H}(\pi)$ obtained by different policies. Error bars correspond to the $90\%$ mean confidence interval. (b) - Distribution of the single-trial objective $F_{1,H}(\pi)$ obtained by different policies. (c) - Mean single-trial objective $F_{1,H}(\pi)$ obtained by the MCTS-based algorithm as a function of the number of expansion steps. Shaded areas correspond to the $90\%$ mean confidence interval. Across all plots, lower is better.

### C.3.9   Adversarial MDP

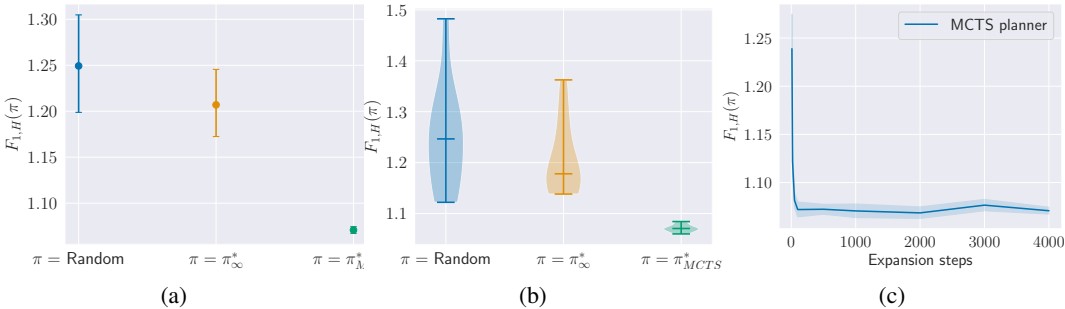

Figure 13: Adversarial MDP: (a) - Mean single-trial objective $F_{1,H}(\pi)$ obtained by different policies. Error bars correspond to the $90\%$ mean confidence interval. (b) - Distribution of the single-trial objective $F_{1,H}(\pi)$ obtained by different policies. (c) - Mean single-trial objective $F_{1,H}(\pi)$ obtained by the MCTS-based algorithm as a function of the number of expansion steps. Shaded areas correspond to the $90\%$ mean confidence interval. Across all plots, lower is better.

