# OpenReview forum: "Solving General-Utility Markov Decision Processes in the Single-Trial Regime with Online Planning"
_ICLR.cc/2026/Conference — ICLR 2026 Poster_

### Official Review · Reviewer_bm8h · 2025-10-21

**Soundness:** 4
**Presentation:** 4
**Contribution:** 2
**Rating:** 8
**Confidence:** 4

**Summary:**

The paper investigates the setting of discounted infinite-horizon general-utility MDPs under the single-trial regime, and proposes a set of theoretical results on the policy class needed to represent the optimal solution, the complexity of achieving such optimum and a methodology to cast one such problem into an equivalent, finite horizon, standard MDP. Finally, an online planning method based on MCTS is proposed, achieving better performances than existing solutions on GUMDP formulation of existing problems.

**Strengths:**

The investigated problem is interesting: standard MDPs are known for being limited in what class of utilities they can represent, and the GUMDP framework allows us to overcome such limitation. However, this setting is much less investigated, and new solution techniques are welcomed. The paper is very clear and rigorous in how it guides the reader through its contributions, always being clear in drawing relevant connections with previously defined concepts and results. As a non-expert in general-utility MDPs, I found it very easy and clear to read. Also, the derived results are sound, and can serve as a good basis for future research in the direction of the single-trial regime. Theory builds up the fundamental pieces until the practical methodology based on MCTS is provided in a clear way.

**Weaknesses:**

I do not have any particular weakness to highlight. The only criticism that can be moved to the current version of the paper is the limited size of the experiment (an aspect the authors have been earnest upfront): the current small experiments are not really telling us much on how the proposed framework would scale to larger setting which are now common in RL (even though the results on the Gym environments are going in this direction a bit). However, I do not feel this is an invalidating weakness, as I appreciate that the nature of the work is mainly theoretical, and the proposed empirical results are still convincing enough to draw interest on this methodology anyway.

**Questions:**

- I think that, at the end of page 2, the symbolism for non-deterministic stationary policies $\Pi^D_S$ and deterministic stationary policies $\Pi_S$ are swapped?

- On line 112, what is $\mathcal{F}$? It is not defined anywhere in the following.

- "*However, the compressed representation used by the occupancy MDP is more amenable to practical implementations since the running occupancy can be incrementally updated as the agent interacts with its environment.*" What do you mean precisely here? That, in practical implementations, you can maintain the histories and incrementally update it with the new "pieces"? Why this could not be done with full-length histories as well in pretty much the same way?

- You say that discounting plays a significant role in your proposed formulation, and reference Proposition 2 as one of those. However, while I do agree that indeed your formulation relies on the fact that you are addressing the infinite-horizon discounted setting, I do not see where discounting is playing a role in such proposition. While, conversely, this is very clear in Proposition 1, that allows you to resort to truncated histories at the price of a constant term (depending on the discount factor $\gamma$) in terms of regret.

- Line 428: GUMDPS $\rightarrow$ GUMDPs

- Results of $\pi_{\text{Solver}}^*$ are a bit surprising: this is using an exact solver in the infinite-trial regime, and as such I would have expected to find a better solution than the method based on MCTS under the single-trial regime (in principle, the solver should be able to always achieve the optimal solution, given enough computational time and resources). Is there any limitation imposed on such solver, like a limited computational time? Could you please discuss on the gap in performance between the optimal solver and your proposed method a bit more?

---

> ### Author Response · Authors · 2025-11-15
> **Response to reviewer bm8h**
>
> We thank the reviewer for the comments. We answer below each of the comments/questions raised.
>
> ### *"I think that, at the end of page 2, the symbolism for non-deterministic stationary policies $\Pi^\textbf{D}\_\text{S}$ and deterministic stationary policies $\Pi^\textbf{D}\_\text{S}$ are swapped?"*
>
> $\Pi^\textbf{D}\_\text{S}$ and $\Pi^\textbf{D}\_\text{S}$ are, indeed, swapped. We fixed this in the new version of the manuscript.
>
> ### *"On line 112, what is $\mathcal{F}$? It is not defined anywhere in the following."*
>
> $\mathcal{F}$ is the sigma algebra of the probability space associated with the trajectories drawn from the (GU)MDP, as defined in [1]. We clarified this in the new version of the manuscript by explicitly stating that $\mathcal{F}$ corresponds to a sigma algebra.
>
> ### *" 'However, the compressed representation used by the occupancy MDP is more amenable to practical implementations since the running occupancy can be incrementally updated as the agent interacts with its environment.' What do you mean precisely here? That, in practical implementations, you can maintain the histories and incrementally update it with the new "pieces"? Why this could not be done with full-length histories as well in pretty much the same way?"*
>
> What we mean is that an agent can keep track of its running occupancy using constant memory (i.e., by simply storing a fixed-size vector), as opposed to keeping track of the entire history, which would require linear space in the number of timesteps. Therefore, the agent can incrementally update its running occupancy as it interacts with its environment.
>
> ### *"You say that discounting plays a significant role in your proposed formulation, and reference Proposition 2 as one of those. However, while I do agree that indeed your formulation relies on the fact that you are addressing the infinite-horizon discounted setting, I do not see where discounting is playing a role in such proposition. While, conversely, this is very clear in Proposition 1, that allows you to resort to truncated histories at the price of a constant term (depending on the discount factor) in terms of regret."*
>
> While it may not be evident in the first place, Proposition 2 relies on discounting to show that there exists a one-to-one mapping between histories and running occupancies (states in the occupancy MDP). To prove such a result, we show in Appendix B.4. that a given mapping (equation (15)) between histories in the single-trial GUMDP and states in the occupancy MDP is a bijection - such a result relies on discounting, not directly carrying to the undiscounted setting. We also agree with the reviewer that discounting plays an important role in Proposition 1.
>
> ### *"Line 428: GUMDPS to GUMDPs"*
>
> We thank the reviewer for spotting this typo. We fixed it in the new version of the manuscript.
>
>
> ### *"Results of are a bit surprising: this is using an exact solver in the infinite-trial regime, and as such I would have expected to find a better solution than the method based on MCTS under the single-trial regime (in principle, the solver should be able to always achieve the optimal solution, given enough computational time and resources). Is there any limitation imposed on such solver, like a limited computational time? Could you please discuss on the gap in performance between the optimal solver and your proposed method a bit more?"*
>
> As discussed in Sec. 4.2, the infinite trials objective, defined in equation (2), differs from the single trial objective (8). Policy $\pi\_\text{Solver}^\*$ is, indeed, the optimal policy for the infinite trials objective. i.e., there are **no computational limitations imposed on the solver** used to find $\pi\_\text{Solver}^\*$. However, in Sec. 5 we focus on evaluating the performance of different policies under the **single-trial** setting. Therefore, $\pi\_\text{Solver}^\*$ underperforms when evaluated under the single-trial setting (since it is optimal for the infinite trials setting and not for the single-trial setting), being outperformed by our method, $\pi\_\text{MCTS}$, which is **explicitly designed to optimize the single-trial** setting. We also emphasize that we are the first work to propose a practical method to solve GUMDPs in the single-trial regime and, hence, there are no other relevant baselines besides $\pi\_\text{Solver}^\*$ to compare against.
>
>
> We hope our answers addressed the reviewer's main concerns.
>
> [1] - Tor Lattimore and Csaba Szepesvari. Bandit Algorithms. Cambridge University Press, 2020. doi:10.1017/9781108571401.

---

> > ### Comment · Reviewer_bm8h · 2025-11-23
> > **Reply to Authors**
> >
> > I would like to thank the authors for their useful rebuttal, that addressed some of my concerns. Given that my score was already positive, I'll keep it as is.

---

### Official Review · Reviewer_LDEZ · 2025-10-27

**Soundness:** 4
**Presentation:** 4
**Contribution:** 3
**Rating:** 8
**Confidence:** 3

**Summary:**

The authors propose a novel method for solving infinite-horizon discounted general utility MDPs in the single-trial regime. This includes some analysis showing how online planning techniques can be used to solve general utility MDPs, as well as empirical results which show that the proposed method yields superior performance when compared to prior methods.

**Strengths:**

The writing quality and presentation of this paper is outstanding. In particular, the analysis performed is well-motivated and presented in a clear and easy-to-read manner. The empirical results are encouraging.

**Weaknesses:**

I have no major concerns with this paper, and it is more or less in a publishable state.

Here are some minor concerns/suggestions for the authors:

- The notation related to the occupancy measure is a bit confusing. In particular, the notation of d_pi in equation 1 vs d_^pi in equation 3 onwards makes it unclear how these two terms are related.
- Lines 173-174: the claim that 'practical applications often require identifying a policy that performs optimally when evaluated based on a single trajectory' is not supported. In particular, the authors need to formally prove/motivate this, or cite a prior work that does so.
- Theorem 1: The use of cost instead of reward may confuse the reader here. In particular, most readers may incorrectly interpret F_1(pi_s) > F_1(pi_m) as indicating that F_1(pi_s) is better than F_1(pi_m) (since most readers are familiar with the reward-based formulation). I would suggest adding some form of clarification in the theorem to avoid this potential issue (or perhaps removing the > altogether and instead only use words).
- I strongly encourage that the authors use the extra page of content allowed during/after the reviews to include more plots related to the empirical results (such as the ones in Appendix C).

**Questions:**

- Did the authors consider whether there are any assumptions needed to ensure that the occupancy measure is well-defined (i.e, that it exists and is unique)? For instance, some MDP-based methods require that the induced Markov chain / MDP has ergodic properties.

---

> ### Author Response · Authors · 2025-11-15
> **Response to reviewer LDEZ**
>
> We thank the reviewer for the comments. We answer below each of the comments/questions raised.
>
> ### *"The notation related to the occupancy measure is a bit confusing (...)"* - on the relation between $d\_\pi$ and $\boldsymbol{\mathrm{d}}^\pi$
>
> We thank the reviewer for the comment. In our work, we define occupancies in two different ways: (i) $d\_\pi$, as defined in equation (1), corresponds to an expected discounted frequency of visitation of state-action pairs; (ii) $\boldsymbol{\mathrm{d}}^\pi$, as introduced in equation (3), corresponds to a random vector encoding the empirical frequency of visitation of state-action pairs associated with the probability space $(\Omega,\mathcal{F}, \mathbb{P}\_\pi)$. In fact, it holds that $d\_\pi = \mathbb{E}[\boldsymbol{\mathrm{d}}^\pi]$. We clarified this in the new version of our manuscript by adding the sentence "It holds that $d\_\pi = \mathbb{E}[\boldsymbol{\mathrm{d}}^\pi]$, for $d\_\pi$ as introduced in (1)".
>
> ### *"Lines 173-174: the claim that 'practical applications often require identifying a policy that performs optimally when evaluated based on a single trajectory' is not supported. In particular, the authors need to formally prove/motivate this, or cite a prior work that does so."*
>
> We added some citations in the new version of our manuscript to further support our claim.
>
> ### *"Theorem 1: The use of cost instead of reward may confuse the reader here. (...) I would suggest adding some form of clarification in the theorem to avoid this potential issue (...)."*
>
> We thank the reviewer for the suggestion. We added a "(lower is better)" text to Theorem 1 to make this clearer in the new version of the manuscript.
>
> ### *"I strongly encourage that the authors use the extra page of content allowed during/after the reviews to include more plots related to the empirical results (such as the ones in Appendix C)"*
>
> We thank the reviewer for the suggestion and will definitely incorporate it in the next version of our manuscript.
>
> ### *"Did the authors consider whether there are any assumptions needed to ensure that the occupancy measure is well-defined (i.e, that it exists and is unique)? For instance, some MDP-based methods require that the induced Markov chain / MDP has ergodic properties."*
>
> We thank the reviewer for the question. In the context of discounted infinite-horizon MDPs/GUMDPs, the discounted occupancies, as introduced in equations (1) and (3), are always well-defined. Therefore, we do not require additional assumptions such as ergodicity to prove our results.
>
> We hope our answers addressed the reviewer's main concerns.

---

> > ### Comment · Reviewer_LDEZ · 2025-11-20
> >
> > I thank the authors for their response to my comments. Given that I did not have major concerns to begin with, I maintain my score.

---

### Official Review · Reviewer_PZ7T · 2025-10-31

**Soundness:** 2
**Presentation:** 1
**Contribution:** 2
**Rating:** 4
**Confidence:** 3

**Summary:**

The paper aims to solve infinite-horizon discounted general-utility Markov decision processes (GUMDPs) in the single-trial (trajectory) evaluation setting. In particular this means that they consider a (non-linear) function $f$ of the empirical discounted occupancy of one trajectory $\mathbb{E}[f(d^{\pi})]$, compared to previous work that considered the expected discounted occupancy $f(\mathbb{E}[d^{\pi}])$. The intuition is that if we consider a non-linear objective and apply standard RL techniques, there is a gap between training and evaluation, since the objective it is being trained on is an upper bound of the true objective.

They achieve this by proving three theorems: (i) non-Markovian policies are necessary for optimality in the single-trial objective; and (ii) reduction of the single-trial problem and solution to a finite-horizon occupancy MDP; and finally (iii) they show the NP-hardness of the policy optimization in the specified regime.

The authors propose to solve the occupancy MDP with MCTS, which guarantees convergence in the infinite iteration regime. Finally the method is tested on illustrative toy and OpenAI Gym environments, showing improvement over baseline random and infinite-trial solver policies. All the proofs and experiments are conducted for discrete state and action spaces.

**Strengths:**

The paper clearly delineates the single-trial from the infinite-trial formulation and establishes a solid method to reformulate the GUMDP as an occupancy MDP that can be solved with well established planning methods. The paper shows clear improvement over previous methods, albeit on simple domains. Overall, the authors identify a subtle problem when considering alternate MDPs with non-linear objectives.

**Weaknesses:**

It is not entirely clear what the main theoretical contribution of the work is, since the single-trial setting is a special case of the multiple-trial setting from Santos et, al. ICML 2025.

Empirically, the method was only shown for very small state and action spaces where even a random policy achieves reasonable performance.  It should ideally be further tested on more complicated and larger environments, and particularly one a specific use case where single-trial evaluation is required by its nature (e.g. stock market predictions, robotics).

The use cases of single-trial evaluation would exclude the existence of a (reliable) simulation. This is a serious limitation of the proposed method that relies on a planner.

**Questions:**

- What is the core difference between GUMDPs considered in this work and Convex MDP? It seems there is no difference if theoretical results follow a convex $f$.
- The random policy already shows decent performance for some of the environments. Could you include harder environments to show the strength of your method?
- Explain the size of the uncertainty in table 1. They seem to vary significantly between environments and policy, the $pi_{MCTS}$ Taxi has even 0 uncertainty, indicating that the uncertainty is most likely underestimated.
- Consider rephrasing lines192-206 on the “Single-trial formulation for GUMDPs” to a list for easier readability.
- Consider using a “hat” the empirical observables to distinguish them clearly from the expected values.
- Mention that you work with discrete state action spaces in the introduction already.
- Introduce examples of use cases of single-trial evaluations.

---

> ### Author Response · Authors · 2025-11-15
> **Response to reviewer PZ7T (part 1)**
>
> We thank the reviewer for the comments and concerns raised. We answer below each of the concerns/questions raised.
>
> ### *"It is not entirely clear what the main theoretical contribution of the work is, since the single-trial setting is a special case of the multiple-trial setting from Santos et, al. ICML 2025."*
>
> The work cited above only studies the case of policy evaluation. In this work, we address the problem of **policy optimization**. Both studies greatly differ since the paper above cited only focuses on providing bounds on the mismatch between the finite and infinite trials formulations for **fixed** (usually non-optimal) **policies**. On the other hand, our work provides results characterizing policy optimization in the single-trial regime for infinite-horizon discounted GUMDPs, investigating which classes of policies suffice for optimality, the hardness of computing optimal policies, as well as how we can leverage online planning techniques to provably solve GUMDPs in the single trial regime in practice.
>
> ### *"Empirically, the method was only shown for very small state and action spaces where even a random policy achieves reasonable performance. It should ideally be further tested on more complicated and larger environments (...)."*
>
> First, we kindly disagree with the reviewer regarding the fact that "the random policy already shows decent performance for some of the environments.". From Table 1, it can be clearly seen that **the random policy is the worst policy** across all tested environments. Furthermore, **our method surpasses the performance of both the random policy and the infinite trials baseline in a statistically significant manner** (for 8 out of the 9 tested experimental settings).
>
> Second, with respect to the chosen environments, we start by highlighting that the GUMDPs framework, as considered in this work as well as in previous works, only accommodates environments with discrete state and action spaces. Having this in mind, we believe **our results to be representative of discrete state-action environments** considered in previous research: we consider three different tasks and four environments **with up to 500 states**. Furthermore, we highlight that we also include mountaincar, an environment that is inherently continuous, but which we discretized and successfully applied our proposed method. We also highlight that no previous works proposed benchmarks for GUMDPs.
>
> Third, the extension of our experiments to environments featuring **continuous** state (or action) spaces is **non-trivial** and would require substantial developments starting right at the formulation of GUMDPs since no previous work has studied the GUMDPs framework under continuously-valued state spaces. Therefore, we believe such a study is out of the scope of our present work. The primary objective of our work is to **establish the foundations of policy optimization for GUMDPs** in the single-trial regime. Our experiments are intended to demonstrate the practical utility of the proposed method, rather than to offer an exhaustive empirical study.
>
> ### *"The use cases of single-trial evaluation would exclude the existence of a (reliable) simulation. This is a serious limitation of the proposed method that relies on a planner."*
>
> We agree with the reviewer on the fact that our proposed method, being based on online planning approaches, relies on the existence of a transition model to simulate trajectories. However, we kindly disagree that this is a major limitation of our method. For several applications the transition model may be available; otherwise, given data collected from the target environment, it can be learned.
>
> ### *"What is the core difference between GUMDPs considered in this work and Convex MDP? It seems there is no difference if theoretical results follow a convex $f$."*
>
> GUMDPs are more general than convex MDPs as they allow the objective function to be non-convex. Our methodology and results apply to any objective function as long as it is Lipschitz (Assumption 1), i.e., we do not require $f$ to be convex. In the experimental section of our work (Sec. 5), the objectives we considered are indeed convex, as we focused on the most common tasks (e.g., exploration and imitation learning), which typically have convex objectives.

---

> ### Author Response · Authors · 2025-11-15
> **Response to reviewer PZ7T (part 2)**
>
> ### Uncertainty estimation for the Taxi environment (Table 1)
> We are confident the uncertainty is well estimated across all our environments, including the Taxi environment. We perform 10 runs per experimental setting and report the 90 percent bootstrapped confidence interval. In Tab. 1, for the Taxi environment under the maximum state entropy exploration task, we reported a value of 0.59 (-0.0, +0.0). While not all runs get exactly the same objective value (as seen in Fig. 7 (b) in Appendix C.3.3.), the MCTS algorithm is able to attain consistently low objective values and, hence, the bounds of the mean confidence interval are indeed very small (hence we report (-0.0, +0.0)). The relatively low variance across all runs is due to the fact that the Taxi environment has deterministic transitions.
>
> ### *"Consider rephrasing lines 192-206 on the "Single-trial formulation for GUMDPs" to a list for easier readability."*
>
> We thank the reviewer for the suggestion and will incorporate it in the next version of our manuscript.
>
> ### *"Consider using a "hat" the empirical observables to distinguish them clearly from the expected values."*
>
> We thank the reviewer for the suggestion and will take it into consideration in the next version of our manuscript. Following a suggestion from other reviewer, we also added the following sentence "It holds that $d_\pi = \mathbb{E}[\boldsymbol{\mathrm{d}}^\pi]$, for $d_\pi$ as introduced in (1)." to the revised manuscript, which we hope better clarifies the connection between the different types of occupancies we consider in our work.
>
> ### *"Mention that you work with discrete state action spaces in the introduction already."*
>
> We thank the reviewer for the suggestion. We added the sentence "We focus our attention to environments with discrete state and action spaces." to our last paragraph in the Introduction.
>
> ### *"Introduce examples of use cases of single-trial evaluations."*
>
> We added some citations to the sentence "This is particularly important, as practical applications often require identifying the policy that performs optimally when evaluated based on a single trajectory of interaction with the environment." (first paragraph of Sec. 2.4) in the revised version of our manuscript.
>
> We hope our answers addressed the reviewer's main concerns.

---

> > ### Comment · Reviewer_PZ7T · 2025-11-27
> > **Author response sufficient**
> >
> > Thank you authors for very thorough response. My main issues are all sufficiently addressed, and I am now on the positive side of evaluating the paper.

---

### Official Review · Reviewer_m7hG · 2025-11-01

**Soundness:** 3
**Presentation:** 3
**Contribution:** 2
**Rating:** 4
**Confidence:** 2

**Summary:**

This paper tackles single-trial optimization in infinite-horizon discounted general-utility MDPs (GUMDPs), where the objective $f$ is possibly a non-linear function of the discounted state-action occupancies. The paper establishes some fundamental results for the problem. It formally proves that non-Markovian policies can strictly outperform stationary and Markovian policies for the single-trial regime. Then the paper introduces the occupancy MDP (OMDP), which is a finite-horizon MDP whose states augment environment states with the running discounted occupancy $o$. The authors establish a one-to-one mapping between histories and OMDP states and characterize equivalence. Based on this connection, they show that solving the OMDP reformulation with stationary policies corresponds to solving the original problem with non-Markovian policies. Despite this, it turns out that policy optimization with smooth convex utility functions is NP-hard.

**Strengths:**

The paper provides some fundamental theoretical results comparing stationary, Markovian, and non-Markovian policies for the single-trial regime. Moreover, the paper characterizes the connection between the original GUMDPs and their truncation to finite-horizon MDPs in terms of regret. The paper also proves that the GUMDPs with non-Markovian policies are equivalent to occupancy MDPs with stationary policies. These theoretical findings are novel and provide interesting insights for the single-trial regime of general utility-based MDP formulations.

**Weaknesses:**

Although the paper empirically tests the computational performance of the MCTS-based algorithmic framework presented in this paper, it lacks its theoretical analysis. The NP-hardness of computing an optimal policy should not prevent us from deriving a finite convergence guarantee to an optimal policy ($1/\epsilon$ versus $\mathrm{log}(1/\epsilon)$). However, there is neither a regret bound nor a convergence guarantee.

**Questions:**

As the authors delineated the connection to occupancy MDPS with stationary policies, would it be possible to design an algorithm with a provable performance guarantee, such as regret bounds and convergence to an optimal policy?

---

> ### Author Response · Authors · 2025-11-15
> **Response to reviewer m7hG**
>
> We thank the reviewer for the comments and concerns raised. We answer below the concerns/questions raised.
>
> ### *"Although the paper empirically tests the computational performance of the MCTS-based algorithmic framework presented in this paper, it lacks its theoretical analysis. The NP-hardness of computing an optimal policy should not prevent us from deriving a finite convergence guarantee to an optimal policy ($1/\epsilon$ versus $\mathrm{log}(1/\epsilon)$). However, there is neither a regret bound nor a convergence guarantee."*:
>
> We thank the reviewer for suggesting the study of regret bounds in the context of solving GUMDPS in the single-trial regime. We agree that NP-Hardness should not preclude us from designing algorithms to come up with approximately optimal policies in practice. This is why, despite our hardness result (Theo. 3), we investigate in Sec. 4 how we can leverage online planning to come up with approximately optimal policies that perform well in practice, as demonstrated by our experiments in Sec. 5.
>
> Regarding the theoretical analysis, we start by highlighting that, in Remark 2 of our submitted manuscript, **we already prove a convergence result** showing that our MCTS-based approach is asymptotically correct. Essentially, Remark 2 states that, as the number of iterations of the MCTS algorithm grows to infinity, we retrieve the optimal policy. Hence, the online planning approach to solve GUMDPs in the single-trial regime successfully asymptotically retrieves the optimal actions for the states encountered as the agent interacts with its environment. Regarding a bound on the **regret** incurred by the MCTS algorithm as a function of the number of expansion steps of the algorithm, we provide a **new remark (Remark 3)** in the new version of the manuscript. This new result states that there exists an MCTS-based algorithm such that, when applied to solve the occupancy MDP with $n$ expansion steps, yields a sequence of policies $(\pi_1, \ldots, \pi_n)$ satisfying $ \mathcal{R}(\pi_1, \ldots, \pi_n) \le O(1/\sqrt{n} + \gamma^H)$. This result follows from our Theo. 2 and the works of [1,2], which establish polynomial regret bounds for MCTS-based algorithms.
>
> Finally, we note that to better incorporate the new Remark 3 in the manuscript we slightly changed our notation, as described in the "general comment" we made for all reviewers above.
>
> ### *"As the authors delineated the connection to occupancy MDPS with stationary policies, would it be possible to design an algorithm with a provable performance guarantee, such as regret bounds and convergence to an optimal policy?"*
>
> Our MCTS-based approach to solve GUMDPs in the single-trial regime comes both with convergence (Remark 2) and regret (Remark 3) guarantees. We refer to our comment above for further details on this matter.
>
>
> We hope our answers addressed the reviewer's main concerns.
>
> [1] - Devavrat Shah, Qiaomin Xie, and Zhi Xu. Non-asymptotic analysis of monte carlo tree search, 2020.
>
> [2] - Can Comer, Jannis Bluml, Cedric Derstroff, and Kristian Kersting. Polynomial regret concentration of UCB for non-deterministic state transitions, 2025.

---

> > ### Comment · Reviewer_m7hG · 2025-11-26
> >
> > Thank you so much for providing the detailed response to the reviewer comments and for expanding the theoretical guarantees on the proposed approach. Based on this, I have increased the score.

---

### Author Response · Authors · 2025-11-15
**General comment to all reviewers**

We thank the reviewers for the questions raised, as well as the positive comments regarding our work. We would like to highlight the following changes, already incorporated into our revised manuscript:

- Following reviewer m7hG's suggestion, we added a new Remark to our manuscript (Remark 3), which provides a bound on the regret incurred by our proposed MCTS to solve GUMDPs in the single-trial regime as a function of the number of expansion steps. The result follows directly by exploiting our Theo. 2 and the works [1,2], which provide polynomial regret bounds for MCTS-based algorithms. Remark 3 complements our previous Remark 2, which analyzes the asymptotic performance of our proposed approach.

- Due to the inclusion of the new Remark 3, and to make the discussion clearer, we slightly changed our notation. More precisely, while in the previous version of our manuscript we referred to $F_{1}(\pi) - \min_{\pi' \in \Pi_{\text{NM}}} F_{1}(\pi')$ as the "regret" and to $F_{1,H}(\pi) - \min_{\pi' \in \Pi_{\text{NM}}} F_{1,H}(\pi')$ as the "truncated regret", we now refer to such quantities as the "optimality gap" and the "truncated optimality gap", respectively. Also, where in the previous version of the article read $\mathcal{R}(\pi)$ and $\mathcal{R}_H(\pi)$, now reads $\textrm{OptGap}(\pi)$ and $\textrm{OptGap}_H(\pi)$. This change in notation aligns with that used in previous works [3] and was necessary to avoid a conflict with the notation introduced in the new Remark 3.

- We made minor corrections and clarifications to the text as suggested by the reviewers.

We answer below the questions and concerns raised by each of the reviewers individually.

[1] - Devavrat Shah, Qiaomin Xie, and Zhi Xu. Non-asymptotic analysis of monte carlo tree search, 2020.

[2] - Can Comer, Jannis Bluml, Cedric Derstroff, and Kristian Kersting. Polynomial regret concentration of UCB for non-deterministic state transitions, 2025.

[3] - Mirco Mutti, Riccardo De Santi, Piersilvio De Bartolomeis, and Marcello Restelli. Convex reinforcement learning in finite trials. JMLR, 24(250):1–42, 2023.

---

### Meta-Review · Area_Chair_jeT1 · 2025-12-22

**Summary:**

**Paper Summary**: he paper aims to solve infinite-horizon discounted general-utility Markov decision processes (GUMDPs) in the single-trial (trajectory) evaluation setting. In particular, they consider a (non-linear) function $f$ of the empirical discounted occupancy of one trajectory $E(f(d_{\pi}))$ , compared to the infinite trial problem $f(E(d_{\pi}))$.  . The intuition is that if we consider a non-linear objective and apply standard RL techniques, there is a gap between training and evaluation, since the objective it is being trained on is an upper bound of the true objective.
They show that: (i) non-Markovian policies are necessary for optimality in the single-trial objective; (ii) the single trial problem can be reduced to an augmented MDP (where MDP is augmented with the occupancy measure); and finally (iii) they show the NP-hardness of the policy optimization in the specified regime.
The authors propose to solve the occupancy MDP with MCTS, which guarantees convergence in the infinite iteration regime. Finally, the method is tested on illustrative toy and OpenAI Gym environments, showing improvement over baseline random and infinite-trial solver policies. All the proofs and experiments are conducted for discrete state and action spaces.

**Reviewers' Concerns**: Reviewers have greatly appreciated the overall contributions of the paper. Some reviewers raised concerns about the contributions, the difference with the convex $f$, and the lack of a provable approximation guarantee of the proposed MCTS-based algorithm.

**AC's take**: The authors' rebuttals have addressed all those concerns. The AC has gone over the paper carefully, and according to the AC, the proposed contributions are solid and will be an asset to the community.

**Reviewer Concerns:**

Reviewers appreciate the contributions. Some of the reviewers raised some minor concerns, which are duly addressed by the authors' rebuttals.

**Reviewer Scores:**

Reviewers' scores were mixed. However, it seems that after the rebuttal, some of the reviewers are willing to increase the score. Hence, they are all aligned with the acceptance decision.

---

### Decision · Program_Chairs · 2026-01-26

Accept (Poster)